# Deep Diffusion-Invariant
# Wasserstein Distributional Classification

**Sung Woo Park**      **Dong Wook Shu**      **Junseok Kwon**
School of Computer Science and Engineering
Chung-Ang University, Seoul, Korea
pswkiki@gmail.com    seowok@naver.com    jskwon@cau.ac.kr

## Abstract

In this paper, we present a novel classification method called deep diffusion-invariant Wasserstein distributional classification (DeepWDC). DeepWDC represents input data and labels as probability measures to address severe perturbations in input data. It can output the optimal label measure in terms of diffusion invariance, where the label measure is stationary over time and becomes equivalent to a Gaussian measure. Furthermore, DeepWDC minimizes the 2-Wasserstein distance between the optimal label measure and Gaussian measure, which reduces the Wasserstein uncertainty. Experimental results demonstrate that DeepWDC can substantially enhance the accuracy of several baseline deterministic classification methods and outperforms state-of-the-art-methods on 2D and 3D data containing various types of perturbations (*e.g.*, rotations, impulse noise, and down-scaling).

## 1   Introduction

The Wasserstein space has been widely used for various machine learning tasks, including generative adversarial learning (2), policy optimization (30), Gaussian processes (17), statistical learning (14), data embedding (6; 19), topic modeling (29), Bayesian inference (1), Gaussian mixture modeling (11), and optimal transport (13; 20; 23). However, the Wasserstein space has not been actively studied for developing novel classification models. In conventional classification problems, $N$ pairs of input data $x_n$ and its corresponding target label $\hat{y}_n$ (*i.e.*, $\{x_n, \hat{y}_n\}_{n=1}^N$) are used for training, where the input data and target label are considered vector-valued points in the Euclidean space, and an objective function is formulated to obtain an inference network $f$ based on the Euclidean distance as follows: $\min_f \sum_n d_E (y_n = f(x_n), \hat{y}_n)$. We aim to answer the following questions in this paper.

**How can the stochastic properties of input data and labels be appropriately captured to handle severe perturbations?**   To answer this question, we represent both input data and target labels as probability measures (*i.e.*, probability densities), denoted as $\mu_n$ and $\hat{\nu}_n$, respectively, in the Wasserstein space and solve a distance-based classification problem (*i.e.*, $\min_f \sum_n \mathcal{W}_2 (\nu_n = f_\#[\mu_n], \hat{\nu}_n)$) based on the 2-Wasserstein distance $\mathcal{W}_2$. Specifically, Euclidean vectors $\{x_n, \hat{y}_n\}$ and inference network $f$ are replaced with *probability measures* $\{\mu_n, \hat{\nu}_n\}_{n=1}^N$ and *push-forward operation* $f_\#[\cdot]$, respectively. Because probability measures can be spread out and multi-modal (19), they provide excellent flexibility for handling stochastic perturbation in data.

Each vector-valued point $x$ is recognized to be sampled from object measure $\mu_n$; $x \sim \mu_n$. Suppose that perturbations are randomly added to data points at test and training time (*i.e.*, $(x + \varepsilon) \sim \mu_n^\varepsilon$ for an unknown random perturbation vector $\varepsilon$ and perturbed probability measure $\mu_n^\varepsilon$). Then, it is reasonable to represent data using a probability measure $\mu_n^\varepsilon$ rather than as an individual *observed data point* $x + \varepsilon$ because data can have multiple locations at every observations (*e.g.*, red points in Fig.1). In this circumstance, we have to minimize the Wasserstein uncertainty of perturbed data represented as probability measures

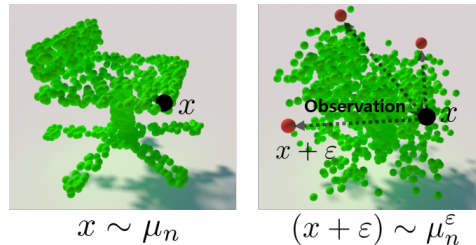

$$x \sim \mu_n \qquad\qquad (x + \varepsilon) \sim \mu_n^\varepsilon$$

Figure 1: **Randomly perturbed data**.

to become predictable. Therefore, the proposed classification method attempts to represent each estimated label stochastically as a probability measure $f_{\#}[\mu_n]$ to consider various types of random perturbations. Such an approach (*i.e.*, considering stochastic data as probability measures in the Wasserstein space) has been studied for stochastic deep neural networks (DNNs) (5). However, stochastic DNNs still represent labels as Euclidean vectors and tend to focus on constructing neural networks rather than theoretical analysis. By contrast, our method represents labels as probability measures and theoretically analyzes classification problems according to these label measures.

**How can classification tasks be defined in the Wasserstein space in an efficient and optimal manner?** It is nontrivial to employ the Wasserstein space for distance-based classification because computing Wasserstein distances directly is intractable. There are two representative approaches for computing Wasserstein distances, which are based on the *primal* and *dual* formulations of optimal transport problems. One approach involves discretizing a data space and finding the discrete optimal coupling of densities that yields the lowest transport cost (7). Linear programming has been adopted to solve the primal problem directly (21; 25). However, the total computational cost grows quadratically as the dimensionality of space and the carnality of data increase. Therefore, this approach cannot be employed for complex data representations. Another approach based on *Kantorovich duality problem* has been widely used in various tasks (2). This approach can represent complex data, but it focuses on the most general case, where probability measures are arbitrary in the Wasserstein space. By contrast, in our method, the Wasserstein distance is defined between a label measure $\nu_n$ and the corresponding fixed Gaussian measure $\mathcal{N}_{\Sigma}$, which enables our method to use a fundamental property of diffusion semi-groups called *hypercontractivity* to impose an explicit upper bound on the 2-Wasserstein distance (*i.e.*, Wasserstein ambiguity) and the exponential decay of the distance.

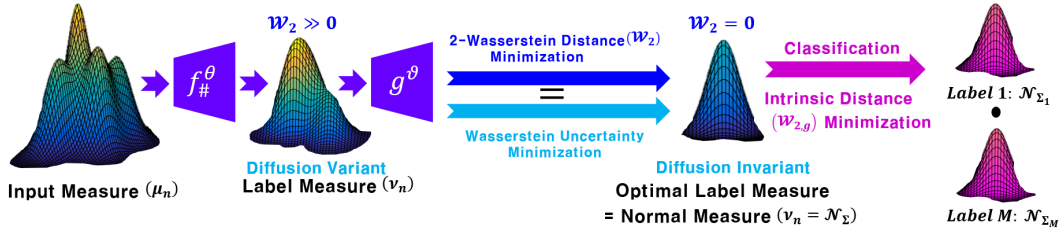

Figure 2: **Deep diffusion-invariant Wasserstein distributional classification (DeepWDC).**

**Proposed method.** Fig.2 presents a conceptual illustration of the proposed method. An input measure $\mu_n$ is fed into the proposed push-forward operator $f_{\#}^{\theta}$, which is parameterized by the neural network $f^{\theta}$, to produce a label measure $\nu_n$. Then, the test function $g^{\vartheta}$ adversarially trained making $\nu_n$ invariant to a pre-defined diffusion operator, which is equivalent to minimizing the Wasserstein uncertainty by tightening the upper bound of $\mathcal{W}_2$. If $\mathcal{W}_2 = 0$, then we obtain the optimal label measure, which is the Gaussian measure $\mathcal{N}_{\Sigma}$. The label of $\mu_n$ is equivalent to that of $\mathcal{N}_{\Sigma_m}$ if the intrinsic distance $\mathcal{W}_{2,g}$ between $\mu_n$ and $\mathcal{N}_{\Sigma_m}$ is optimally minimized in the Wasserstein Gaussian subspace. Thus, our method aims to find the optimal parameters $\theta$ and $\vartheta$ for two DNNs. Our contributions can be summarized as follows:

• We introduce a novel distance-based distributional classification method (**DeepWDC**), where both input data and target labels are *probability measures* in the 2-Wasserstein space. To make classification problems computationally tractable, we indirectly derive 2-Wasserstein distances by determining their upper bounds (*i.e.*, Wasserstein ambiguity). We present theoretical evidence supporting the capability of our method to rapidly reduce Wasserstein ambiguity.

• We prove that minimizing Wasserstein ambiguity is equivalent to making $\nu_n$ (*i.e.*, the estimated label measure) *diffusion-invariant*. If $\nu_n$ is diffusion-invariant, then the density of $\nu_n$ is stationary over time. During theoretical analysis, we introduce the concept of *hypercontractivity of a diffusion semi-group* to relate diffusion invariance to the Wasserstein distance.

• We experimentally demonstrate the robustness of our method against severe random perturbations (*e.g.*, rotations, downscaling, and non-homogeneous local noises) and verify that it can substantially outperform state-of-the-art deterministic methods.

**Wasserstein Gaussian embedding.** Muzellec and Cuturi explored the Gaussian 2-Wasserstein space (19), where embedded points are represented as non-degenerate Gaussian measures. Because a Bures metric is explicitly defined in this space, no sub-routines for computing this metric are required.

However, this method focuses on point embedding in the Gaussian Wasserstein subspace rather than the distributional realization of the feature space. Therefore, solving classification problems is problematic in this space because the actual density of objects cannot be represented as a unique elliptic distribution. By contrast, our method imposes an explicit constraint that necessarily transforms inference measures into Gaussian measures, which are uniquely defined.

**Wasserstein distributional learning.** In recent studies, data uncertainty has been represented as distance in the Wasserstein space (8; 21; 25; 26; 9). To this end, the Wasserstein distance was employed to define the uncertainty of data, which was referred to as the Wasserstein ambiguity set (mathematically equivalent to a Wasserstein ball). The uncertainty of data was measured by considering the closeness of the induced probability density of the data to the prescribed target density in terms of the distance metric. Our method can be interpreted within this framework, where the radius of the Wasserstein ambiguity set is proportional to the square root term in (2). Our method minimizes this Wasserstein ambiguity for classification tasks to handle severe perturbations from a distributional perspective.

## 2 The Proposed Method

In this section, we describe the two fundamental concepts of our method, namely, *diffusion invariance* and *Wasserstein Gaussian subspaces*. Then, we define an objective function based on these concepts for classification tasks in the Wasserstein space. Finally, we present a Wasserstein-distance-based classifier.

As mentioned in Section 1, the $n$-th input data sample and its target label are represented as probability measures $\mu_n$ and $\hat{\nu}_n$, respectively. The label measure $\nu_n$ is estimated using a push-forward operation $f_\#[\mu_n]$ (*i.e.*, $\nu_n = f_\#[\mu_n]$). Then, an input vector $x$ can be sampled from $\mu_n$ (*i.e.*, $x \sim \mu_n$), and the estimated label measures are denoted as $\nu_n$. We define the test function $g$ as an element of the class of smooth real-valued functions with compact support (*i.e.*, $g \in C_0^\infty$), where $g$ maps $y$ to a real value (*i.e.*, $g(y) \in \mathbb{R}$). In this paper, $f_\#[\mu_n]$ and $g(y)$ are implemented as DNNs with parameters $\theta$ and $\vartheta$ denoted as $f_\#^\theta[\mu_n]$ and $g^\vartheta(y)$, respectively.

### 2.1 Diffusion-Invariant Measure

Our method imposes a diffusion operator on probability measures to derive diffusion-invariant measures of the estimated label $\nu_n$ and computes the corresponding Wasserstein ambiguity (*i.e.*, the upper bound of the 2-Wasserstein distance $\mathcal{W}_2$).

**Definition 1.** *(Diffusion Operator) Given a Markov semi-group $P_t$ at time $t$, the diffusion operator (i.e., infinitesimal generator) $\mathcal{L}$ of $P_t$ is defined as*

$$\mathcal{L}g(y) = \lim_{t \to 0} \frac{1}{t} \left( P_t g(y) - g(y) \right) = \sum_{i,j} \frac{\partial^2}{\partial y_i \partial y_j} B^{ij}(y) g(y) - \sum_i A^i(y) \frac{\partial}{\partial y_i} g(y), \quad (1)$$

*where $B$ and $A$ are the matrix and vector-valued measurable functions, respectively, $B^{ij}$ denotes the $(i,j)$-th function of $B$, and $A^i$ denotes the $i$-th component function of $A$.*

The diffusion operator $\mathcal{L}g$ can be considered the average change in $g$ with respect to an infinitesimal change in time according to the Markov semi-group. If the expectation of the average change in $g$ is zero, then the data sampled from the probability measure $y \sim \nu_n$ are considered to be stationary over time. In this case, $\nu_n$ becomes invariant to $\mathcal{L}$ (*i.e.*, diffusion invariant) (please refer to Section 3.3 for details regarding perturbation analysis).

**Definition 2.** *(Diffusion-invariant Measure) Given the diffusion operator $\mathcal{L}$, the probability measure $\nu_n$ is considered to be invariant to $\mathcal{L}$, when $\mathbb{E}_{y \sim \nu}[\mathcal{L}g(y)] = 0$ for any $g \in C_0^\infty$.*

We set a target label measure $\hat{\nu}_n$ to $\mathcal{N}_{\Sigma_n}$ (*i.e.*, a centered non-degenerate Gaussian measure with covariance $\Sigma_n$). This setting is possible because $\mathcal{N}_{\Sigma_n}$ is an element of the 2-Wasserstein space. Additionally, we set $B^{ij}$ and $A^i$ in (1) to $\Sigma_{ij}$ (*i.e.*, the $(i,j)$-th entry of $\Sigma$) and $y^i$ (*i.e.*, the $i$-th component of $y$), respectively. Then, the Wasserstein ambiguity is represented by $\mathcal{L}$ as follows:

**Proposition 1.** *Let $\mathcal{L}g^\vartheta(y)$ is diffusion operator defined in Definition 1. Then,*

$$\mathcal{W}_2(f_\#^\theta[\mu_n], \hat{\nu}_n) = \mathcal{W}_2(f_\#^\theta[\mu_n], \mathcal{N}_{\Sigma_n}) \leq \sqrt{\sup_\vartheta \mathbb{E}_{y \sim f_\#^\theta[\mu_n]}[|\mathcal{L}g^\vartheta(y)|]}. \quad (2)$$

As shown in (2), we can minimize the 2-Wasserstein distance (*i.e.*, $\mathcal{W}_2(f_{\#}^{\theta}[\mu_n], \hat{\nu}_n) = 0$) by minimizing its upper bound (*i.e.*, $\sup_{\vartheta} \mathbb{E}_{y \sim \nu_n} \left[ \mathcal{L}g^{\vartheta}(y) \right] = 0$), meaning that the estimated label measure $\nu_n$ becomes diffusion-invariant, as defined in Definition 2, and is hardly affected by perturbations. We can make $\sup_{\vartheta} \mathbb{E}_{y \sim \nu_n} \left[ \mathcal{L}g^{\vartheta}(y) \right]$ equal to zero if and only if $f_{\#}^{\theta}[\mu_n] = \mathcal{N}_{\Sigma_n}$. Thus, the goal of our method is to make $\nu_n = f_{\#}^{\theta}[\mu_n]$ similar to $\mathcal{N}_{\Sigma_n}$ (*i.e.*, $\nu_n = \mathcal{N}_{\Sigma_n}$) by updating the parameter $\theta$. In (2), the upper bound is minimized at a square root rate and can rapidly converge to zero. Section 3.1 presents the proof of Proposition 1 and further investigations.

The diffusion operator $\mathcal{L}$ in (1) contains a second-order derivative term. Therefore, it is inefficient for a neural network $g$ to calculate a Hessian matrix $\nabla_y^2 g(y)$ during training. However, our method can calculate the diffusion operator without any derivatives. With respect to the proposed diffusion operator $\mathcal{L}$ with the specific settings $B^{ij} = \Sigma_{ij}$ and $A^i = y^i$, the Markov semi-group $P_t g$ has an explicit form called *Mehler's formula*: $P_t g^{\vartheta}(y) = \mathbb{E}_{Z \sim \mathcal{N}_{\mathbf{I}}} \left[ g^{\vartheta} \left( e^{-t} y + \sqrt{1 - e^{-2t}} \Sigma_n^{\frac{1}{2}} Z \right) \right]$, where $\mathcal{N}_{\mathbf{I}}$ denotes the standard Gaussian measure. Then, the diffusion operator is calculated as

$$
\begin{aligned}
\mathbb{E}_{y \sim \nu_n}[\mathcal{L}g^{\vartheta}(y)] &= \lim_{t \to 0} \mathbb{E}_{y \sim f_{\#}^{\theta}[\mu_n]} \left[ \frac{P_t g^{\vartheta}(y) - g^{\vartheta}(y)}{t} \right] \\
&= \lim_{t \to 0} \frac{1}{t} \mathbb{E}_{y \sim f_{\#}^{\theta}[\mu_n]} \mathbb{E}_{Z \sim \mathcal{N}_{\mathbf{I}}} \left[ g^{\vartheta} \left( e^{-t} y + \sqrt{1 - e^{-2t}} \Sigma_n^{\frac{1}{2}} Z \right) - g^{\vartheta}(y) \right].
\end{aligned}
\tag{3}
$$

Our method has several mathematical advantages over existing methods. For example, the WGAN-GP method (2) adopts a gradient penalty term for its test function to induce 1-Lipschitzness. However, this method involves a strong global penalty for the test function. Such a penalty is inevitably caused by the assumption of the Kantorovich–Rubinstein formula. However, the test function $g$ in the proposed method only uses the local properties of Markov semi-groups with no assumptions regarding prior conditions.

## 2.2 Wasserstein Gaussian Subspaces

We define *Wasserstein Gaussian subspaces* to compute the intrinsic distance between the estimated label measure $\nu_n$ and target label measure $\hat{\nu}_n$.

**Definition 3.** *The Wasserstein Gaussian subspace $\mathcal{P}_{2,g}$ is a subspace of the 2-Wasserstein space $\mathcal{P}_2$, which consists of centered non-degenerate Gaussian measures, where $\mathcal{W}_{2,g}$ is a distance metric in this space.*

Suppose that each estimated label vector is mapped to the hypersphere, $y_n = f(x_n)$ and $\hat{y}_n \in \mathbb{S}^{d-1} \subset \mathbb{R}^d$. In such cases, $d_{E=\mathbb{R}^d}$ is not the same as $d_{S=\mathbb{S}^{d-1}}$, and we cannot use $d_{E=\mathbb{R}^d}$ as a true metric for minimizing the distance between the estimated and target labels. Therefore, the intrinsic distance $d_{\mathbb{S}^{d-1}}$ must be defined to derive accurate objective functions for classification tasks. To define the intrinsic distance, we introduce the Wasserstein Gaussian subspace (*i.e.*, hypersphere), where the target label measure $\hat{\nu}_n$ is represented as a Gaussian measure $\mathcal{N}_{\Sigma_n}$ and the estimated label measure $\nu_n$ also lies within this subspace. Then, the intrinsic distance is defined as $\sum_{n=1}^{N} \mathcal{W}_{2,g}(\nu_n, \hat{\nu}_n) = \sum_{n=1}^{N} \mathcal{W}_{2,g}(f_{\#}[\mu_n], \mathcal{N}_{\Sigma_n})$. In Section 3.2, we analyze the geometric characteristics of the Wasserstein Gaussian subspace.

## 2.3 Objective Function

The proposed objective function is defined as follows:

$$
\min_{\theta} \max_{\vartheta} \frac{1}{N} \sum_{n=1}^{N} \underbrace{\mathbb{E}_{y_n \sim f_{\#}^{\theta}[\mu_n]} [|\mathcal{L}g^{\vartheta}(y_n)|]}_{\text{Diffusion invariance term}} + \underbrace{\mathcal{W}_{2,g}(f_{\#}^{\theta}[\mu_n], \mathcal{N}_{\Sigma_n})}_{\text{Intrinsic distance term}}.
\tag{4}
$$

The first term of (4), which is defined in (3), ensures the *diffusion invariance* of the estimated label measure $\nu_n$, where $\nu_n$ can be invariant to diffusion operators (Section 2.1). This term also determines the Wasserstein ambiguity and yields the upper bound for the 2-Wasserstein distance in (2). The second term of (4), which is defined in (6), minimizes the *intrinsic distance* between the estimated label measure $\nu_n$ and target label measure $\mathcal{N}_{\Sigma_n}$ in the Wasserstein Gaussian subspace (Section 2.2). This term reduces infinite-dimensional problems in the Wasserstein space to finite-dimensional tractable problems, which can easily optimized by $f$ and $g$. Our method aims to find the optimal parameters $\theta$ and $\vartheta$ for two neural networks $f$ and $g$, respectively, that minimize the objective function in (4). Therefore, our estimated label measures are close to the target label measures and are robust to stochastic perturbations. In the Supplementary Materials, algorithms are provided to outline the entire procedure of the proposed method.

## 2.4 Evaluation Metric

To evaluate the classification performance, we propose the following classifier based on 2-Wasserstein distance. We calculate the Top-1 average accuracy for $N$ objects as:

$$\hat{\mathbf{cls}}(\{\mu_n\}_{n=1}^N) = \frac{1}{N} \sum_{n=1}^{N} \mathbf{cls}\left[\mu_n, \underset{\mathcal{N}_{\Sigma_m^*}}{\arg\min} \, \mathcal{W}_{2,g}\left(f_\#^\theta[\mu_n], \mathcal{N}_{\Sigma_m}\right)\right], \tag{5}$$

where $\mathbf{cls}[\mu_n, \mathcal{N}_{\Sigma_m^*}] = 1$ if $\mathcal{N}_{\Sigma_m^*}$ and $\mu_n$ share the same label information. We search for the probability measure of target label $\mathcal{N}_{\Sigma_m^*}$ that is the closest to the probability measure of estimated label $f_\#^\theta[\mu_n]$. Then, input data $\mu_n$ is classified into the label of $\mathcal{N}_{\Sigma_m^*}$. If we can find the optimal parameters $(\theta^*, \vartheta^*)$ for the neural networks $(f, g)$, then $\hat{\mathbf{cls}}(\{\mu_n\}_{n=0}^{N-1})$ in (5) is 1. Thus, we can solve the metric-based classification problem in the Wasserstein space.

## 3 Theoretical Analysis

In Section 2, we present the basic concepts of the proposed method. In this section, we decompose each term in (4) by examining the connection between the diffusion operator and 2-Wasserstein distance, and we detail the theoretical advantages of the proposed method. We first introduce the main assumptions used in this theoretical analysis subsequently.

Because our method aims to make each push-forward measure $\nu_n$ diffusion invariant with respect to $\mathcal{L}$, we assume that there is a sequence $\{\delta_{k,n}\}_{k=0}^\infty$ such that $\delta_{k,n} = \sup_{g \in C_0^\infty} \int |\mathcal{L}g(y)| \, q_{k,n} d\mathcal{N}_{\Sigma_n}(y) < \infty$, where $q_{k,n}$ denotes the density of $\nu_{k,n}$, and there is a large $K_0 \in \mathbb{N}^+$ that satisfies $\delta_{k,n} = 0$ if $k \geq K_0$ and $\forall n = 0, \cdots, N$. Intuitively, $\delta_{k,n}$ can be understood as a indicator that how $n$-th label measure $\nu_{k,n}$ is diffused according to $\mathcal{L}$ at learning iteration $k$.

The Supplementary Materials include other minor assumptions with notations and the details of the entire theoretical analysis, which cannot be presented herein owing to space constraints.

### 3.1 Hypercontractivity

We examine the relationship between the proposed diffusion operator $\mathcal{L}$ (defined in Section 2.1) and the 2-Wasserstein distance $\mathcal{W}_2$. For simplicity, the subscript $n$ (*e.g.*, in $\nu_n$) is omitted in this section.

**Proposition 2.** *(Descending of the Wasserstein Ambiguity Set) Let $\{\nu_k\}_{k=0}^\infty$ be a sequence of probability measures satisfying assumptions. Then, $\nu_k \in B_{\mathcal{W}_2}(\mathcal{N}_\Sigma, \sqrt{\delta_k})$. In other words, $\mathcal{W}_2(\nu_k, \mathcal{N}_\Sigma) \leq \sqrt{\delta_k}$. Furthermore, $\mathcal{W}_2(\nu_k, \mathcal{N}_\Sigma) \to 0$ as $k \to \infty$.*

This proposition is inspired by the *Bakry-Emery inequality* (3) and *HWI-inequality* (22), which describe the fundamental behavior of the Markov diffusion semi-group. To prove Proposition 2, Fisher information is replaced with the proposed diffusion term $\sup_\vartheta \mathbb{E}_{y \sim \nu_k} \left[|\mathcal{L}g^\vartheta(y)|\right]$ in (2), which clearly highlights the connection between the diffusion operator $\mathcal{L}$ and the 2-Wasserstein distance. Specifically, *the 2-Wasserstein distance is bounded above by the square root of the diffusion term.* Fig.3 presents a conceptual illustration of Proposition 2. The inequality in Proposition 1 is different form that in Proposition 2.

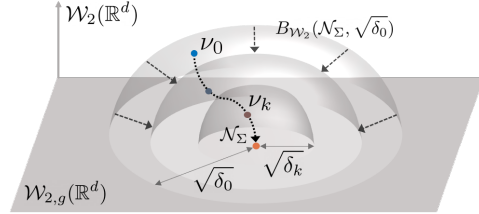

Figure 3: **Descending of the Wasserstein Ambiguity Set.** The radius of the ambiguity set converges to zero (*i.e.*, $r = \sqrt{\delta_k} \to 0$).

**Proposition 3.** *(Exponential Decay of Wasserstein Distance) We define a sub-sequence $\tau(k) \subset \mathbb{N}^+$, such that $\tau(k) = \{k'; \frac{1}{k'} \left(\int \zeta d[\nu_{k'} - \nu_0] + \epsilon(k')\right) \leq \delta_0, k \leq k'\}$, where $\zeta \in C_b$, and let $|\epsilon(k)|$ be a dual error satisfying $|\epsilon(k)| \to 0$ as $k \to \infty$. In this case, the following inequality holds: $\mathcal{W}_2(\nu_{\tau_k}, \mathcal{N}_\Sigma) \leq \sqrt{\delta_0 e^{-2\tau_k} + \epsilon(\tau_k)}$, where $\tau_k$ denotes an element of $\tau(k)$.*

By setting $\zeta(y)$ to a 1-Lipschitz function, the inequality in Proposition 3 can be rewritten according to its definition and the Kantorovich-Rubinstein formula as follows: $\frac{1}{k} \left(\mathcal{W}_1(\nu_k, \nu_0) + \varepsilon(k)\right) \leq \delta_0$. Therefore, if the average difference in terms of 1-Wasserstein distance between the initial and updated measures (*i.e.*, $\nu_{\tau_k} = f_\#^{\theta_{\tau_k}}[\mu]$) is bounded above by $\delta_0$, the Wasserstein ambiguity (*i.e.*, radius of 2-Wasserstein balls) *exponentially decays to zero.* Therefore, with a small error of $\epsilon$ and a moderate update rate of $f^\theta$, the Wasserstein ambiguity decays exponentially.

## 3.2 Riemannian Geometry of $\mathcal{W}_{2,g}$

We interpret the proposed Wasserstein Gaussian subspaces $\mathcal{W}_{2,g}$ (defined in Section 2.2) as *totally geodesic submanifolds* in the 2-Wasserstein space analogous to the convex set (*e.g.*, a unit cube with one-hot vectors as elements). In this totally geodesic submanifold, any geodesic that is calculated using induced Riemannian metrics is also defined in the ambient manifold. Therefore, if the Wasserstein Gaussian subspace is a totally geodesic submanifold in the 2-Wasserstein space, then $\mathcal{W}_2$ and $\mathcal{W}_{2,g}$ coincide and have an explicit form (*i.e.*, the *Bures metric*) defined as follows:

**Definition 4.** *Let $\Sigma_n$ be defined as $\Sigma_n = \mathbf{Cov}(X_n), X_n \sim \nu_n$. Then, the Bures metric between probability measures $\nu_n$ and $\tilde{\nu}_n$ is defined as*

$$d_B(\nu_n, \tilde{\nu}_n) = \sqrt{\mathbf{Tr}\left(\Sigma_n + \tilde{\Sigma}_n - 2\left(\Sigma_n^{\frac{1}{2}}\tilde{\Sigma}_n\Sigma_n^{\frac{1}{2}}\right)^{\frac{1}{2}}\right)}. \tag{6}$$

**Proposition 4.** *(4; 18) The Wasserstein Gaussian subspace $\mathcal{P}_{2,g}(\mathbb{R}^d)$ is a totally geodesic submanifold in the 2-Wasserstein space $\mathcal{P}_2(\mathbb{R}^d)$. Thus, the following distances are equivalent: $d_B(\nu_n, \tilde{\nu}_n) = \mathcal{W}_{2,g}(\nu_n, \tilde{\nu}_n) = \mathcal{W}_2(\nu_n, \tilde{\nu}_n)$ for any $\nu_n, \tilde{\nu}_n \in \mathcal{P}_{2,g}(\mathbb{R}^d)$.*

In (4), we assume that the label measure $\nu_n$ is Gaussian, which is generally not true at the training time, and we approximate the intrinsic distance $\mathcal{W}_{2,g}$ to $d_B$. However, according to Propositions 2 and 4 as well as (2), $d_B$ can be an exact estimation of $\mathcal{W}_{2,g}$ if the optimal parameters $(\theta, \vartheta)$ of the neural networks $(f, g)$ are found and $\nu_n$ lies in the Wasserstein Gaussian subspace. Using the explicit form presented in Definition 4, we can calculate approximated 2-Wasserstein distances during testing.

## 3.3 Perturbation Analysis

Let $\mu$ be an input data measure and $\mu^\varepsilon$ be a perturbed measure induced by unknown random perturbations. The corresponding label measures for $\mu$ and $\mu^\varepsilon$ are defined as $\nu = f_\#^\theta[\mu]$ and $\nu^\varepsilon = f_\#^\theta[\mu^\varepsilon]$, respectively. It is nontrivial to construct an explicit form of the perturbation function due to the complexity of neural network. Therefore, we assume that the mass of $\nu$ is transported along $(\mathbf{Id} + \varepsilon h)$. Then, the perturbed measure $\nu^\varepsilon$ can be written as follows:

$$\nu^\varepsilon = (\mathbf{Id} + \varepsilon h)_\#[\nu], \ \varepsilon > 0, \ \varepsilon \sim p_\varepsilon(y), \tag{7}$$

where $h$ is a *perturbation function* defined on the feature space with a magnitude $\varepsilon \sim p_\varepsilon$, and the corresponding information is assumed to be unknown at both the training and testing times. In this context, we answer two crucial questions: First, how is the average perturbed distance $\mathbb{E}_\varepsilon \mathcal{W}_{2,g}(\nu_k^\varepsilon, \mathcal{N}_\Sigma)$ related to the diffusion operator $\mathcal{L}$ or $\delta_k$? Second, what are the theoretical advantages of the proposed method over conventional deterministic models? The following propositions answer these questions.

**Proposition 5.** *(Wasserstein Perturbation) Let $\nu_k^\varepsilon = (\mathbf{Id} + \varepsilon h)_\#\nu_k$ be a perturbed measure by $\varepsilon h$. Then, there exist numerical constants $0 \leq \kappa_1, \kappa_2 < \infty$ such that mean radius of perturbed Wasserstein ambiguity set $\nu_k^\varepsilon \in B_{\mathcal{W}_2}(\mathcal{N}_\Sigma, r')$ is bounded as:*

$$\mathbb{E}_\varepsilon \mathcal{W}_2(\nu_k^\varepsilon, \mathcal{N}_\Sigma) \leq \sqrt{d\kappa_1\kappa_2\mathbb{E}[\varepsilon] + \delta_k}. \tag{8}$$

Consider a deterministic model in which label measures are considered as Dirac-delta measures in the feature space, meaning $\nu = \delta_y, \nu^\varepsilon = \delta_{y+\varepsilon h}$, and the target label measure is considered $\hat{\nu} = \delta_z$. In this case, the Wasserstein distance is equivalent to the Euclidean distance and has a deterministic upper bound defined as $\mathbb{E}_\varepsilon \mathcal{W}_2(\nu^\varepsilon, \delta_z) = \mathbb{E}_\varepsilon d_E(y + \varepsilon h, z) \leq \|h\|_2 \mathbb{E}[\varepsilon] + d_E(y, z)$. For simplicity, assume that $d\kappa_1\kappa_2 \leq 1$ in (8) and $\|h\|_2 \leq 1$. Then, the Wasserstein uncertainty definitions for the deterministic and stochastic models are given by

$$\underbrace{\mathbb{E}[\varepsilon] + d_E(y, z)}_{\text{Deterministic Model}} \Longleftrightarrow \underbrace{\sqrt{\mathbb{E}[\varepsilon] + \delta_k}}_{\text{Stochastic Model}}. \tag{9}$$

There are two major implications of (9) that verify the theoretical advantages of our method over conventional deterministic models. First, it is clear that the proposed method efficiently reduces the randomness of label information by considering $\delta_k$ in (9), whereas deterministic models are incapable of handling the stochastic properties of perturbations. Second, the Wasserstein ambiguity is proportional to the square root of $\mathbb{E}[\varepsilon]$ in our stochastic model, which ensures a minor impact to the model in the presence of large perturbations (*i.e.*, $\mathbb{E}[\varepsilon] \gg 1$). To show the effectiveness of diffusion term to classification accuracy, we assume the binary classification task with perturbed label measure $\nu^\varepsilon$ where the positive and negative target label measures are denoted by $\mathcal{N}_{\Sigma_+}$ and $\mathcal{N}_{\Sigma_-}$.

**Corollary 1.** *(**Perturbed Binary Classification**.) Let $\Sigma_+$ and $\Sigma_-$ be a $r$-rank SPD matrices, and $\varepsilon \sim p_\varepsilon = \exp(b)$ be an exponential distribution with parameter $b$. Then, the probability of $\nu^\varepsilon$ classified as positive labels is bounded as follows:*

$$\mathbb{P}[\mathbf{cls}(\nu^\varepsilon) = 1] \leq 1 - e^{-\frac{br(\lambda_{max}^+ + \lambda_{max}^-) - 4b\delta}{4d\kappa_1\kappa_2}}, \tag{10}$$

*where $\lambda_{\max}^+$ and $\lambda_{\max}^-$ denote maximum eigenvalues of matrices $\Sigma_+$ and $\Sigma_-$, respectively.*

Corollary 1 shows that the probability of correct classification is maximized with the exponential ratio, if we can find the optimal parameters $(\theta, \vartheta)$ to attain $\delta_k \approx 0$. In Proposition 6, we consider an extreme case in which $Y_{t=k}$ is a stochastic process that enforces the path for the Markov semi-group $P_k$ (defined in Section 2.1) for which the corresponding law is a label measure $\nu_k = f_\#^{\theta_k}[\mu]$. In other words, each particle exactly follows the Ornstein-Uhlenbeck process, which is known to have an explicit path. This proposition verifies that the probability of the average norm of the perturbation function $\nu_k(\mathbb{E} \|h\|_2^2)$ can be efficiently minimized.

**Proposition 6.** *(**Markov Inequality for the Perturbation Function**) Let $Y_k \sim \nu_k$ denote the Markov-process related to the Markov semi-group and its corresponding law $\nu_k$. For the $l$-th component of the perturbation function $h_l \in \mathbf{L}^1(\nu_k)$, we denote $T(y) = \|h(y)\|_2^2 < \infty$. Then, there are numerical constants $0 \leq \kappa_3, \kappa_4 < \infty$ such that*

$$\nu_k\big(\mathbb{E}_y[T(Y_k)] \geq a\big) \leq \frac{\kappa_3}{a^2} e^{\frac{1}{2(e^{2k}-1)} + 2\varepsilon^2} \left(d\kappa_4 + k\delta_k\right), \tag{11}$$

*for $y \in \mathbb{R}^d$. Furthermore, $\lim_{k \to \infty} \nu_k(\mathbb{E}[T(Y_k)] \geq a) \to a^{-2} e^{2\varepsilon^2} d\kappa_3\kappa_4$.*

# 4 Experiments

For empirical validation, we applied our method to two classification problems: 3D point cloud classification using the ModelNet10 (28) dataset and image classification using the CIFAR10 dataset (12), where the data suffer from various perturbations. There are different considerations in this work that can be compared to those in previous works.

• We generated severe *structural perturbations*, including random rotations, random resizing, and non-homogeneous local noises. These perturbations are different from those considered in previous works, such as the generalization of adversarial examples, which simply add noise to original images and construct the maximum bound in terms of $L_p$-distance in the pixel space.

• We randomly changed the data representation at the *training* and *testing* times. This setting aims to model real-world situations in which unknown severe random perturbations potentially exist in data.

The Supplementary Materials elucidate the specifications of the **network architectures, perturbation setup and samples, additional experiments, and more ablation studies**.

## 4.1 Implementation Details

**Covariance Matrices**. Each target Gaussian measures are represented as $r$-rank degenerate covariance matrices, $\Sigma_c = \mathbf{M}_c^T \mathbf{M}_c$, where $\mathbf{M}_c$ denotes $(r \times d)$ size of random matrix for the $c$-th class and all indices are i.i.d uniform random variables. For our 2D image classification experiments, we used $\mathbf{Sym}_{128}^+$ of rank-32 covariance matrices to represent centered Gaussian measures. For our 3D image classification experiments, we used $\mathbf{Sym}_{128}^+$ rank-3 covariance matrices. To calculate the square roots of the covariance matrices efficiently, we used the GPU-friendly Newton Schulz algorithm presented in (15). Because computational complexity increases quadratically even using this algorithm, we limited the maximum number of dimensions of the Gaussian measures to $\mathbf{dim}(\mathcal{P}_{2,g}) \leq d(d+1)/2 \approx 8K$, where $d = 128$.

**Hyperparameters and Training Setup**. We used the ADAM optimizer with a learning rate of $10^{-5}$ for the network $g$ and a learning rate of $10^{-3}$ for the network $f$, as well as for the baseline networks. All experiments were executed using a single RTX 2080 TI GPU. Our method was implemented using `Pytorch1.4.0` and `Python3.6`.

**Algorithm**. Algorithm 1 outlines the entire procedure of the proposed method.

## 4.2 Comparison with Deterministic Models

**2D Image Classification.** If the spatial information of 2D images is highly distorted by severe perturbations, then the accuracy of conventional classification networks decreases significantly. In

---

**Algorithm 1** DeepWDC

---

**Require:** Neural networks $f, g$ with initial weights $\theta, \vartheta$.

Initialize covariance matrices $\{\Sigma_n\}_{n=1}^N$ with rank-$s$ and sample $x_n \sim \mu_n$.

**for** $k = 1$ to $K$ (*i.e.*, the total number of training iterations) **do**

   1) Optimize the diffusion invariance term in (4).

$$\nabla_{\theta_k^1} \leftarrow \nabla_{\theta_k} \mathbb{E}_{x_n, Z \sim \mathcal{N}_\mathbf{I}, t \sim U[0,1]} \frac{1}{\Delta t} \left[ \left\| g\left( e^{-\Delta t} f^{\theta_k}(x_n) + \sqrt{1 - e^{-2\Delta t}} \Sigma_n^{\frac{1}{2}} Z \right) - g\left( f^{\theta_k}(x_n) \right) \right\| \right].$$

   2) Estimate the sample covariance matrices.

$$\tilde{\Sigma}_n(\theta_k) \leftarrow \frac{1}{d-1} \left[ f^{\theta_k}(x_n) - \mathbb{E} f^{\theta_k}(x_n) \right]^T \left[ f^{\theta_k}(x_n) - \mathbb{E} f^{\theta_k}(x_n) \right].$$

   3) Optimize the intrinsic distance term in (4).

$$\nabla_{\theta_k^2} \leftarrow \nabla_{\theta_k} \sqrt{\mathbf{Tr}\left( \Sigma_n + \tilde{\Sigma}_n(\theta_k) - 2\left( \Sigma_n^{\frac{1}{2}} \tilde{\Sigma}_n(\theta_k) \Sigma_n^{\frac{1}{2}} \right)^{\frac{1}{2}} \right)}.$$

   4) Update $f$ by descending its stochastic gradients $\nabla_{\theta_k^1} + \nabla_{\theta_k^2}$.

$$\nabla_{\vartheta_k} \leftarrow \nabla_{\vartheta_k} \mathbb{E}_{x_n, Z \sim \mathcal{N}_\mathbf{I}, t \sim U[0,1]} \frac{1}{\Delta t} \left[ \left\| g^{\vartheta_k}\left( e^{-\Delta t} f(x_n) + \sqrt{1 - e^{-2\Delta t}} \Sigma_n^{\frac{1}{2}} Z \right) - g^{\vartheta_k}\left( f(x_n) \right) \right\| \right].$$

   5) Update $g$ by ascending its stochastic gradients $\nabla_{\vartheta_k}$.

**end for**

---

contrast, our method delivers accurate classification results under these conditions. 2D convolutions are highly vulnerable to rotations and local perturbations, because information integration is dependent on the spatial structures of 2D images. For example, each pixel is integrated by convolutional kernels in which the connectivity between the pixels is conditioned (*i.e.*, 2D grid). However, our method does not require such pre-conditioning because it uses the holistic (distributional) information of group of pixels in feature space, which is recognized as a form of high dimensional push-forward probability density. Therefore, our method can generate robust representations in convolutional feature spaces even in the presence of severe structural perturbations.

Table 1: **2D image classification accuracy (in %) on the CIFAR10 dataset with perturbations.** Regarding perturbation types and amounts, we set parameters for impulse noise ($\epsilon$), rotation ($\theta, \theta_2$), random scaling ($sc$), random shearing ($sh$), and random crop ($cc$). For the deterministic perturbation, we used CIFAR10-C (10) benchmark dataset. "#Param." denotes the number of parameters in the network. The best results are presented in **bold**.

| Methods (#Param.) | $\{e\}$ | $\{\theta, sc, sh, \epsilon\}$ | $\{\theta_2, sc, sh, \epsilon\}$ | $\{\theta_2, cc\}$ | $\{\theta_2, cc_2\}$ | CIFAR10-C |
|---|---|---|---|---|---|---|
| ResNet ($11.1M$) | 89.4 | 73.6 | 73.4 | 76.9 | 77.9 | - |
| DenseNet ($6.8M$) | 86.6 | 76.3 | 75.9 | 78.9 | 74.8 | 92.6 |
| DeepWDC ($6.7M$) | **95.9** | **88.6** | **93.0** | **92.7** | **87.7** | **95.8** |

**3D Point Cloud Classification.** For 3D point cloud classification, we considered DGCNN (27) and PointNet++ (24) as baselines. We compared our method with the baselines and the results are summarized in Table 2. As shown in the table, the conventional networks are highly vulnerable to geometric perturbations and jitters with high variance because such perturbations induced drastic changes in features. Therefore, accuracy decreased significantly as additional perturbation types are applied. By contrast, our method exhibits significantly smaller drops in accuracy in the presence of severe perturbations.

Table 2: **3D point cloud classification accuracy (in %) for the ModelNet10 dataset with perturbations.** For perturbations, we set parameters for random sampling ($T$), random scaling ($s$), random jitter ($\epsilon$), and random rotation ($\theta$). The best results are presented in **bold**.

| Methods (#Param) | $\{T\}$ | $\{T, s, \epsilon\}$ | $\{T, s, \epsilon, m\}$ | $\{T, s, \epsilon_2\}$ | $\{T, s, \epsilon_3\}$ | $\{T, s, \epsilon_3, \theta\}$ |
|---|---|---|---|---|---|---|
| PointNet$^{++}$ ($1.7M$) | 96.6 | 71.7 | 67.3 | 47.2 | 35.7 | 25.8 |
| DGCNN ($1.9M$) | **97.1** | 83.6 | 79.1 | 68.0 | 54.8 | 35.4 |
| DeepWDC ($0.96M$) | 96.9 | **94.8** | **91.0** | **85.3** | **71.9** | **55.2** |

### 4.3   Ablation Study

We analyzed DeepWDC by evaluating each of its component. First, we examined the role of the diffusion invariance term in (4). As shown in Fig.4(a), DeepWDC without the diffusion invariance term delivers low accuracy with increased variance, whereas using $\mathbb{E}[\mathcal{L}g] + d_{2,g}^{\mathcal{W}}$ yields stable and accurate results. We can interpret the proposed diffusion invariance term as *distributional regularization*, where a group of pixels in the feature space is forced to match a Gaussian density. There is a canonical isomorphism between the Wasserstein Gaussian subspace and positive semi-

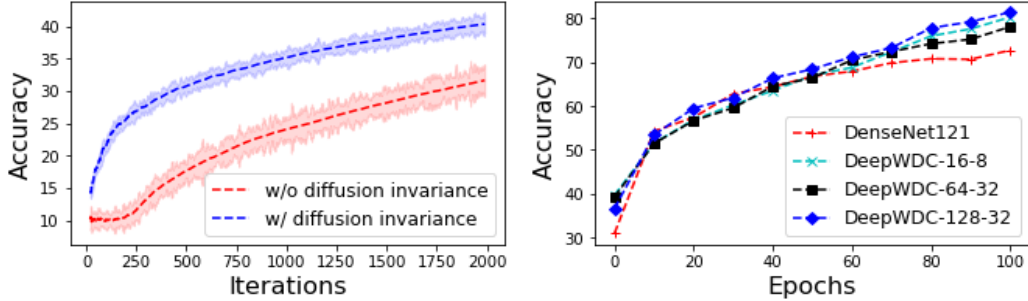

(a) Effectiveness of the diffusion invariance term      (b) Classification results with different dimensionality

Figure 4: **Effectiveness of the proposed DeepWDC**. (a) The blue curve represents the results of using both diffusion invariance $\mathbb{E}[\mathcal{L}g]$ and intrinsic distance terms $\mathcal{W}_{2,g}$ in (4), whereas the red curve represents using the intrinsic distance term alone. The plot depicts the average accuracy with $m \pm 0.2\sigma$. (b) Curves show classification results with different dimensionality of the proposed DeepWDC. Both experiments are conducted on 2D image classification tasks on CIFAR10.

definite matrix space (16) (*i.e.*, $\mathcal{P}_{2,g}(\mathbb{R}^d) \cong \mathbf{Sym}_d^+$). Thus, the dimensionality of the covariance matrices is another key factor for accurate classification. We set up an experiment by testing different dimensions of $d = 16, 64, 128$ and ranks $r = 8, 32$. As shown in Fig.4(b), setting dimension $\mathbf{dim}(\mathcal{P}_{2,g}) = d(d+1)/2 \approx 8K$, $d = 128$ with rank-32 label measures produced the best result.

## 5   Conclusion

In this paper, we proposed a novel classification method, called DeepWDC, where input data and labels are represented as probability measures to address severe perturbations in input data. DeepWDC can output the optimal label measure in terms of diffusion invariance, where the label measures are stationary and become equivalent to Gaussian measures. Experimental results verified that DeepWDC significantly outperforms state-of-the-art classification methods on both 2D and 3D data in the presence of various types of perturbations.

## Acknowledgements

This work was supported by Institute for Information & communications Technology Planning & evaluation (IITP) grant funded by the Korea government (MSIP) (No.2017-0-01780).

## Broader Impact

The proposed framework can considerably enhance conventional classification methods, of which performance is very sensitive to various types of perturbations (*e.g.*, rotations, impulse noise, and down-scaling). The proposed Wasserstein distributional classifier represents both input data and target labels as probability measures and its diffusion invariant property prevents the classifier from being affected by severe perturbations. Hence, various research fields under real-world environments can benefit from exploiting our framework to obtain accurate classification results.

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
