[Supplementary Material]

# Supplementary Materials for Deep Diffusion-Invariant Wasserstein Distributional Classification

## 1 Mathematical Backgrounds

### 1.1 Markov Diffusion Semi-group

In this section, we introduce definitions and propositions that are used for proofs hereafter.

**Definition 1.** *The Markov semigroup $(P_t)_{t \geq 0}$ in $\mathbb{R}^d$ acting on a function $f \in C_0^\infty$ is defined as follows:*

$$P_t f(x) = \int f(x') p_t(x, dx'), \tag{1}$$

*where $p_t(x, dx')$ is called a transition kernel and is a probability measure for all $x$ and $t \geq 0$.*

**Definition 2.** *(Diffusion Operator) Given a Markov semi-group $P_t$ at time $t$, the diffusion operator (i.e., infinitesimal generator) $\mathcal{L}$ of $P_t$ is defined as*

$$\mathcal{L}g(y) = \lim_{t \to 0} \frac{1}{t} \left( P_t g(y) - g(y) \right) = \sum_{i,j} \frac{\partial^2}{\partial y_i \partial y_j} B^{ij}(y) g(y) - \sum_i A^i(y) \frac{\partial}{\partial y_i} g(y), \tag{2}$$

*where $B$ and $A$ are matrix and vector-valued measurable functions, respectively; $B^{ij}$ denotes the $(i, j)$-th function of $B$; $A^i$ denotes the $i$-th component function of $A$.*

**Definition 3.** *(Diffusion carre du champ) Let $f, g \in C_0^\infty$. Then, we define a bilinear form $\Gamma_c$ in $C_0^\infty \times C_0^\infty$ as*

$$\Gamma_c(f, g) = \frac{1}{2}[\mathcal{L}\Gamma_{c-1}(fg) - \Gamma_{c-1}(f\mathcal{L}g) - \Gamma_{c-1}(g\mathcal{L}f)], \tag{3}$$

*for $c \geq 1$.*

We denote $\Gamma(f) \equiv \Gamma(f, f)$. The bilinear form $\Gamma$ can be considered a generalization of the integration by parts formula, where $\int f\mathcal{L}g + \Gamma(f)d\xi = 0$ for the invariant measure $\xi$ of $\mathcal{L}$.

**Definition 4.** *(Relative Fisher Information) For $\Gamma$ defined earlier, the relative Fisher information of $\nu_t$ with respect to $\mathcal{N}_\Sigma$ is defined as $I(\nu_t | \mathcal{N}_\Sigma) = \int P_t[q]^{-1} \Gamma(P_t[q]) d\mathcal{N}_\Sigma$.*

**Definition 5.** *(Curvature-Dimension condition) We say that infinitesimal generator $\mathcal{L}$ induces the $CD(\rho, \infty)$ curvature-dimension condition if it satisfies $\Gamma_1(f) \leq \rho\Gamma_2(f)$ for all $f \in C_0^\infty$.*

**Example. (Multivariate Gaussian distribution)**

Because our diffusion operator generates an Ornstein-Uhlenbeck semi-group, the curvature-dimension condition can be explicitly calculated. Through simple calculations, the first order $(c = 1)$ diffusion carré du champ is induced follows:

$$\Gamma_1(f) = \left( [\nabla f]^T \Sigma \nabla f \right)^2. \tag{4}$$

Similarly, the second order $(c = 2)$ diffusion carré du champ is calculated as follows:

$$\begin{aligned}
\Gamma_2(f) &= \frac{1}{2} \left[ \mathcal{L} \left( \Gamma_1(f^2) \right) - 2\Gamma_1 \left( f, \mathcal{L}(f) \right) \right] \\
&= \mathbf{Tr} \left( \left[ \Sigma \nabla^2 f \right]^2 \right) + \left( [\nabla f]^T \Sigma \nabla f \right)^2 = \mathbf{Tr} \left( \left[ \Sigma \nabla^2 f \right]^2 \right) + \Gamma_1(f),
\end{aligned} \tag{5}$$

for arbitrary $f \in C_0^\infty$. While $\mathbf{Tr}\left(\left[\Sigma\nabla^2 f\right]^2\right)$ is non-negative, $\Gamma_1 \le \Gamma_2$, and our diffusion operator $\mathcal{L}$ induces the $CD(1,\infty)$ curvature-dimension condition. Therefore, we always set $\rho = 1$ in this paper. In [4], other types of diffusion operators are considered including Gamma distributions.

**Proposition 1.** *(Decay of Fisher information along a Markov semigroup [1]) If we assume the curvature-dimension condition $CD(\rho,\infty)$, then $I(\nu_{k,t}|\mathcal{N}_\Sigma) \le e^{-2\rho t}I(\nu_k|\mathcal{N}_\Sigma)$.*

Proposition 1 demonstrates the exponential decay of Fisher information if the curvature-dimension condition $CD(1,\infty)$ is satisfied. This property will play a central role in proving Proposition 2.

## 1.2   Riemannian Diffusion $\mathcal{L}_g$

In the previous section, we discussed the connection between the diffusion operator and Wasserstein distance in Proposition 2 and examined the basic and well-known properties of Markov semi-groups. In this section, we investigate the geometric implications of $\mathcal{L}$. Let $(\mathbb{R}^d, G)$ denote a Riemannian manifold equipped with a flat (constant) metric tensor $G$, which is defined as the inverse of the covariance matrix $\Sigma \in \mathbf{Sym}_+^d$. We consider the Riemannian diffusion $\mathcal{L}_g$ for this manifold as follows:

$$\mathcal{L}_g = \Delta_g - \nabla_g \log(p_\Sigma), \tag{6}$$

where $\Delta_g$ and $\nabla_g$ denote the Laplace-Beltrami operator and Riemannian gradient with a flat metric tensor $G = \Sigma$, respectively. We define the probability measure $\nu$ as absolutely continuous with respect to the Riemannian Lebesgue measure $\nu \ll \mathbf{m}_g$ with a density $p_\Sigma(x) = e^{-\mathbf{U}(x)}$ such that $d\nu = p_\Sigma d\mathbf{m}_g$, where $U$ is twice differentiable. Because our metric tensor is flat, we can easily calculate the Riemannian diffusion operator as follows:

$$
\begin{aligned}
\mathcal{L}_g f &= \Delta_g f + \nabla_g \log(p_\Sigma) f \\
&= (\mathbf{det}(G))^{-\frac{1}{2}} \sum_{i,j} \partial_i \left((\mathbf{det}(G))^{\frac{1}{2}} g^{ij} \partial_j f\right) + g^{ij} \partial_i \log(p_\Sigma) \partial_j f \\
&= \sum_{i,j} g^{ij} \partial_i \partial_j f + g^{ij} \frac{\nabla p_\Sigma}{p_\Sigma} \partial_j f = \sum_{i,j} \Sigma_{ij} \partial_i \partial_j f + \sum_j y_j \partial_j f \\
&= \mathbf{Tr}(\Sigma \nabla^2 f) + X^T \nabla f = \mathcal{L}f, \quad f \in C_0^\infty(\mathbb{R}^d),
\end{aligned}
\tag{7}
$$

where we denote $\{g^{ij} g_{ik}\}_{i,k} = G^{-1}G = \Sigma^{-1}\Sigma = \mathbf{I}$. The final form is equivalent to (1) in the main paper, which implies that the distance in the feature space can be recognized as flat Riemannian distance (*i.e.*, *Mahalanobis distance*).

## 2   Notations and Assumptions

The $l$-th component of function $g$ is denoted as $g_l : \mathbb{R}^d \to \mathbb{R}$. $C_b(\mathbb{R}^d)$ denotes the set of continuous and bounded functions in $\mathbb{R}^d$, and $C_0^\infty(\mathbb{R}^d)$ denotes the set of $\infty$-class functions with compact support in $\mathbb{R}^d$. The $L_\alpha$-norm of function $f \in \mathbf{L}_\alpha(\nu)$ is denoted as $\|f\|_{\alpha,\nu} = \left(\int |f|^\alpha d\nu\right)^{\frac{1}{\alpha}}$. The Euclidean $L_\alpha$-norm of vector $v \in \mathbb{R}^d$ is denoted as $\|v\|_\alpha$. Let $dy$ be a Lebesgue measure and $\mathcal{N}_\Sigma$ be a non-degenerate centered Gaussian measure with density $p_\Sigma(x) = \left(2\pi^{-\frac{d}{2}}\mathbf{det}(\Sigma)^{-1}\right) e^{-\frac{1}{2}x^T \Sigma^{-1} x}$, where $d\mathcal{N}_\Sigma = p_\Sigma dx$ and $\Sigma$ is a $(d \times d)$ covariance matrix. The $n$-th label measure at the $k$-th iteration is denoted as $\nu_{k,n} = f_\#^{\theta_k}[\mu_n]$.

**Definition 6.** *(2-Wasserstein space).*   *Let $\mathcal{P}(\mathbb{R}^d)$ be a space of probability measures. Then, its subspace, $\nu \in P_2(\mathbb{R}^d)$, which satisfies $\mathbf{m}_2(\nu) = \int_\Omega d(y_0, y)^2 d\nu < \infty$ for every $y_0 \in \mathbb{R}^d$, is called the 2-Wasserstein space.*

According to the definition, we can get the following inclusion: $P(\mathbb{R}^d) \subset P_2(\mathbb{R}^d) \subset P_{2,g}(\mathbb{R}^d)$. In the 2-Wasserstein space, a metric ball is denoted as $B_{\mathcal{W}_2}(\mathcal{N}_\Sigma, r) = \left\{\nu; d_2^W(\nu, \mathcal{N}_\Sigma) \le r\right\}$ with radius $r$ centered at $\mathcal{N}_\Sigma$.

We make the following assumptions:

- **H1.** Each parameterized push-forward measure $\nu_n = f_\#^\theta[\mu_n]$ is absolutely continuous with respect to the Gaussian measure $\mathcal{N}_{\Sigma_n}$, ($i.e., \nu_n \ll \mathcal{N}_{\Sigma_n}$). Therefore, each measure $\nu_n$ has a density $q_n$ such that $d\nu_n = q_n d\mathcal{N}_{\Sigma_n}$, the existence of which is assured by the Radon-Nikodym theorem.

- **H2.** There is a monotone-decreasing sequence $\{\delta_{k,n}\}_{k=1}^\infty$ ($i.e., \delta_{k,n} \geq \delta_{k+1,n}, \forall k \in \mathbb{N}^+$) such that $\delta_{k,n} = \sup_{g \in C_0^\infty} \int |\mathcal{L}g(y)| q_{k,n} d\mathcal{N}_{\Sigma_n}(y) < \infty$, and a large constant $K_0 \in \mathbb{N}^+$ satisfying $\delta_{k,n} = 0$, if $k \geq K_0, \forall n = 0, \cdots, N$, where $q_{k,n}$ denotes the density of the $n$-th label measure $\nu_{k,n}$ at the $k$-th sequence.

- **H3.** Each inference (push-forward) measure $\nu_{k,n} = q_{k,n} d\mathcal{N}_{\Sigma_n}$ at the $k$-th sequence is centered ($i.e., \mathbb{E}_{y \sim \nu_{k,n}}[y] = 0$).

- **H4.** The second central moment of push-forward measures are bounded as $\mathbf{m}_2(\nu_{k,n}) = \int_{\mathbb{R}^d} d(y_0, y)^2 d\nu_{k,n} < \infty$ for every $y_0 \in \mathbb{R}^d$. In other words, $\nu_{k,n}$ is assume to be an element of $\mathcal{P}_2(\mathbb{R}^d)$ according to the definition of the 2-Wasserstein space.

The second assumption **H2** implies that we can always find an optimal solution for the proposed objective function, when there is an index $K_0$ such that $\delta_{K_0}$ converges to zero. For our inference network $f$, we normalize every convolutional features such that $\mathbb{E}[f^{\theta_k}(x_n)] = 0$ to satisfy **H3**. The other mild assumptions are required for proofs.

**Remark**. It should be noted if $\mu_n$ and $\hat{\nu}_n$ ($i.e.$, target measures) are Dirac-delta measures ($i.e., \mu_n = \delta_{x_n}, \hat{\nu}_n = \delta_{\hat{y}_n}$), then the corresponding Wasserstein distance $\mathcal{W}_2(f_\#[\mu_n], \hat{\nu}_n)$ is equivalent to $d_E(f(x_n), \hat{y}_n)$. Therefore, our method can be considered a generalized version of conventional distance-based classification models.

## 2.1 Neural Network As a Member of $C_0^\infty$

In this paper, the proposed neural network $g$ is assumed to be a member of $C_0^\infty$, which seems unnatural because of the non-differentiability of the neural network for a finite number of points where the non-differentiability is induced by activation units such as ReLU. However, it can be easily shown that there is always some smooth function $g_s \in C_0^\infty$ such that $\|g_s - g\| \approx 0$. Let a perturbed dataset lie in a compact subset $\Omega \subset \mathbb{R}^{\hat{d}}$, and let the inference network as $f : \Omega \subset \mathbb{R}^{\hat{d}} \to \mathbb{R}^d$ map perturbed data points into a $d$-dimensional feature space and $\mathrm{supp}(f) = \Omega$. To guarantee the smoothness of neural network $g$ in a global sense, we convolve it with a mollifier function $\psi$. First, we introduce the function:

$$\psi_\epsilon(y) = \frac{1}{C_\epsilon \epsilon} \psi(y/\epsilon), \tag{8}$$

where $\psi$ is a standard mollifier function $\psi(y) = e^{-\frac{1}{1-\|y\|_2^2}}$ with $\epsilon > 0$ and the constant $C_\epsilon = \frac{1}{\int \psi_\epsilon d\nu}$.

For the next step, we define a compactly supported neural network $\hat{g}$ as follows:

$$\hat{g}(y) = \begin{cases} \|y\|_{2,\nu}^2 & \|y\|_{2,\nu}^2 \leq M = \sup_Z |Z|_{2,\nu}^2 \\ 0 & \|y\|_{2,\nu}^2 > M, \end{cases} \tag{9}$$

where $Z \in f(\Omega)$ is a perturbed random vector related to the probability measure $f_\#[\mu]$. Then, $\hat{g}$ is compactly supported by an $\mathbf{L}_2$-ball with radius $M$, because perturbed points lie in some compact set $\Omega \subset \mathbb{R}^{\hat{d}}$ and $Z \in f(\Omega)$ is also compact in $\mathbb{R}^d$. Finally, we take convolution $\psi_\epsilon$ in (8) with $\hat{g}$ in (9) to induce smoothness of the neural network $g$ as follows:

$$g_s(y) = [\hat{g} * \psi_\epsilon](y) = \int \hat{g}(y)\psi_\epsilon(y - y')d\nu(y'). \tag{10}$$

The smoothness of $g$, ($i.e., g \in C_0^\infty$) can be easily verified, because $\partial_i g = \hat{g} * \partial_i \psi_\epsilon, \forall i$. By the Young's inequality, the following inequality holds:

$$\|g_s\|_{1,\nu} = \|\hat{g} * \psi_\epsilon\|_{1,\nu} \leq \|\hat{g}\|_{1,\nu} \|\psi_\epsilon\|_{1,\nu} = \|\hat{g}\|_{1,\nu}, \tag{11}$$

where $\|\hat{g} - \hat{g} * \psi_\epsilon\|_{1,\nu}$ converges to zero, as $\epsilon$ approaches zero. Let $B(M)$ be an $\mathbf{L}_2$-ball with radius $M$. Then, $g|_{B(M)} = \hat{g}|_{B(M)} \approx \hat{g} * \psi_\epsilon|_{B(M)} = g_s|_{B(M)}$ for a sufficiently small $\epsilon$, regardless of $f$.

**Remark.** In the analysis presented in [2], mathematical assumptions required for neural networks are discussed. Similarly, this section is designed to discuss and deliver the proper mathematical setting for the assumption used in the main paper (*i.e.*, a neural network is a member of $C_0^\infty$).

## 3 Proofs

In this section, we prove the propositions introduced in the main paper.

### 3.1 Proof of Proposition 2

**Proposition 2.** *(Descending of the Wasserstein Ambiguity Set) Let $\{\nu_k\}$ be a sequence of probability measures satisfying assumptions. Then, $\nu_k \in B_{\mathcal{W}_2}(\mathcal{N}_\Sigma, \delta_k^{\frac{1}{2}})$, which is equivalent to $\mathcal{W}_2(\nu_k, \mathcal{N}_\Sigma) \leq \delta_k^{\frac{1}{2}}$ and $\mathcal{W}_2(\nu_k, \mathcal{N}_\Sigma) \to 0$ as $k \to \infty$.*

*Proof.* We define the Markov semi-group $P_t$ for the auxiliary variable $t$, where $\lim_{t\to 0+} \mathbb{E}_{\nu_k}[P_t f] = \mathbb{E}_{\nu_k}[f]$. We assume that $\nu_{k,t}$ is absolutely continuous with respect to the centered Gaussian measure $\mathcal{N}_\Sigma$ (*i.e.*, $\nu_t \ll \mathcal{N}_\Sigma$). Based on this assumption, we let $q_{k,t}$ be the corresponding density (*i.e.*, $d\nu_{k,t} = q_{k,t}d\mathcal{N}_\Sigma$). Then, the subscript $t$ is interpreted as the path index of the probability measure from $\nu_{k,t=0}$ to $\nu_{k,t}$, where the path is defined as the solution of the continuity equation in a distributional sense as follows:

$$\partial_t \rho_t = \nabla \cdot (\rho_t v_t), \quad v_t = \nabla \log q_{k,t}. \tag{12}$$

Because the density of the Gaussian measure $\mathcal{N}_\Sigma$ has the form of $\frac{1}{Z}e^{-\eta(x)}$, where $\eta(x) = \frac{1}{2}x^T\Sigma^{-1}x, \nabla^2\eta(x) \geq 0$ and $Z$ is a normalization constant, we can simply use the results presented in [5]. By integrating both sides of the inequality in (Lemma 2, [5]) with respect to the auxiliary variable $t \in (0, \infty)$, we can derive the following inequality:

$$\mathcal{W}_2(\nu_k, \mathcal{N}_\Sigma) = \int_0^\infty \frac{d}{dt_+} \mathcal{W}_2(\nu_k, \nu_{k,t})dt \leq \int_0^\infty \sqrt{I(\nu_{k,t}|\mathcal{N}_\Sigma)}dt. \tag{13}$$

In the inequality above, we replace the Fisher information with the diffusion generator $\mathcal{L}$ as follows:

$$\mathcal{W}_2(\nu_k, \mathcal{N}_\Sigma) \leq \int_0^\infty \sqrt{I(\nu_{k,t}|\mathcal{N}_\Sigma)}dt$$
$$= \int_0^\infty \sqrt{\int P_t^{-1}q_k \Gamma(P_t q_k) d\mathcal{N}_\Sigma}dt = \int_0^\infty \sqrt{\int \mathcal{L}(-\log P_t q_k) d\nu_{k,t}}dt. \tag{14}$$

The second equality above is derived using the property of the bilinear operator $\Gamma$ with respect to diffusion operator $\mathcal{L}$ which is defined as follows:

$$\int P_t^{-1}q_k \Gamma(P_t q_k) d\mathcal{N}_\Sigma = -\int \mathcal{L}(\log P_t q_k)q_k d\mathcal{N}_\Sigma = \int \mathcal{L}(-\log P_t q_k)d\nu_{k,t} \geq 0. \tag{15}$$

We denote $|g| = g^+$ for simplicity. According to Proposition 1, we can relate $\nu_{k,t}$ to its initial term $\nu_{k,t=0}$ as follows:

$$\int_0^\infty \sqrt{\int \mathcal{L}(-\log P_t q_k) d\nu_{k,t}}dt \leq \int_0^\infty \sqrt{e^{-2\rho t}\int \mathcal{L}(-\log P_0 q_k)\, d\nu_{k,t=0}}dt$$
$$\leq \int_0^\infty \sqrt{e^{-2\rho t}\sup_{g\in C_0^\infty}\int \mathcal{L}^+ g q_k d\mathcal{N}_\Sigma}dt$$
$$= \int_0^\infty \sqrt{e^{-2\rho t}}dt\underbrace{\sqrt{\sup_{g\in C_0^\infty}\int \mathcal{L}^+ g d\nu_k}}_{\text{Diffusion invariance term}} = \rho^{-1}\delta_k^{\frac{1}{2}}. \tag{16}$$

The second inequality is naturally induced, because the proposed objective function is defined to select the maximal elements over the set of functions $g \in C_0^\infty$, and $\mathcal{L}g \leq \mathcal{L}^+ g$. $\qquad\square$

**Remark.** In this paper, we only focused the invariance property of diffusion-type Fokker-Planck equation and their diffusion semi-group (Ornstein-Uhlenbeck) which induces curvature-dimension constant to be $\rho = 1$. Thus, the geometric-characteristics of diffusion semi-group is not highlighted in the main paper. But the original results developed in the paper [5] can be generalized to our method. Specifically, the target measure $\hat{\nu}$ in our method can be generalized to much general form such as a Gibbs measure having density $d\hat{\nu} = \frac{1}{Z}e^{-\Psi(x)}dx$ for arbitrary smooth $\Psi$. In this case, the diffusion semi-group can be designed to induce larger curvature-dimension constant $\rho \gg 1$ by considering other types of $\Psi$ to give finer radius of Wasserstein ambiguity set for classification.

## 3.2 Proof of Proposition 3

**Proposition 3.** *(Exponential Decay of Wasserstein Distance)* Let us define the sub-sequence $\tau(k) \subset \mathbb{N}^+$ such that $\tau(k) = \left\{ k' \big| \frac{1}{k'} \left( \int \zeta d[\nu_{k'} - \nu_0] + \epsilon(k') \right) \leq \delta_0, k \leq k' \right\}$, where $\zeta \in C_b$, and let $|\epsilon(k)|$ be a dual error satisfying $|\epsilon(k)| \to 0$ as $k \to \infty$. In this case, the following inequality holds: $d_2^{\mathcal{W}}(\nu_{\tau_k}, \mathcal{N}_\Sigma) \leq \sqrt{\delta_0 e^{-2\tau_k} + \epsilon(\tau_k)}$, where $\tau_k$ denotes the element of $\tau(k)$.

*Proof.* It should be note that $\tau(k) \subseteq \tau(k')$ for any $k \leq k'$ because we define $\delta_k$ as monotonically decreasing. This statement is trivial if we show the inequality.

The proof consists of three steps. First, we derive the inequality $\mathbb{E}_{\nu_k}\zeta \leq \mathbb{E}_{\xi_k}\zeta + \epsilon(k)$ from the assumptions for $\nu_0$ defined under sub-sequence conditions.

**First step:** Let $\xi_k$ be another auxiliary probability measure such that $\xi_k \ll \mathcal{N}_\Sigma$ and $d\xi_k = P_k q_0 d\mathcal{N}_\Sigma$, and let $\mathcal{L}^+\zeta = |\mathcal{L}\zeta|$. Then, the inequality $\mathbb{E}_{\nu_k}\zeta \leq \mathbb{E}_{\xi_k}\zeta + \epsilon(k)$ is written as follows:

$$
\begin{aligned}
\int \zeta q_k d\mathcal{N}_\Sigma &\leq \int \zeta P_k q_0 d\mathcal{N}_\Sigma + \epsilon(k) = \int P_k \zeta q_0 d\mathcal{N}_\Sigma + \epsilon(k) \\
&\leq \int \zeta q_0 d\mathcal{N}_\Sigma + \int \int_0^k P_s \mathcal{L}^+\zeta(y) q_0(y) ds d\mathcal{N}_\Sigma(y) + \epsilon(k) \\
&= \int \zeta d\nu_0 + \int_0^k \left[ \int P_s \mathcal{L}^+\zeta(y) d\nu_k(y) \right] ds + \epsilon(k).
\end{aligned} \tag{17}
$$

The first equality is induced by the properties of diffusion-type Markov semi-groups (*i.e.*, $\int [\zeta(y)P_k q_0 - q_0 P_k \zeta(y)] d\mathcal{N}_\Sigma(y) = 0$). Because $\mathcal{L}^+\zeta$ is non-negative and measurable, we can apply Tonelli's theorem to the second line, which creates an inequality. According to assumption **H2**, we can derive the following inequality:

$$
\int P_s \mathcal{L}^+\zeta(y) = \int \underbrace{\left[ \int \mathcal{L}^+\zeta \left( e^{-s}y + \sqrt{1 - e^{-2s}}\Sigma^{\frac{1}{2}}z \right) d\nu_0(y) \right]}_{\leq \sup_g \int \mathcal{L}^+ g d\nu_0 = \delta_0} d\mathcal{N}_\Sigma(z) \leq \delta_0. \tag{18}
$$

Therefore, we obtain the following inequality:

$$
\mathbb{E}_{\nu_k}[\zeta] - \mathbb{E}_{\nu_0}[\zeta] - \epsilon(k) \leq k\delta_0. \tag{19}
$$

By rearranging both sides of this expression and rescaling the error term $\epsilon(k) \to -\frac{\epsilon(k)}{k}$[1], we obtain the following inequality:

$$
\frac{1}{k} \int \zeta d[\nu_k - \nu_0] = \frac{1}{k} \int \zeta(q_k - q_0) d\mathcal{N}_\Sigma + \epsilon(k) \leq \delta_0. \tag{20}
$$

Therefore, $\mathbb{E}_{\nu_k}\zeta \leq \mathbb{E}_{\xi_k}\zeta + \epsilon(k)$ is equivalent to the assumptions for $\nu_{\tau_k}, \tau_k \in \tau(k)$.

**Second step:** We use the Kantorovich duality of the Hopf-Lax semi-group to induce the inequality of the Wasserstein distances. Prior to presenting the second step of the proof, we introduce the following proposition.

**Proposition A.** *(Kantorovich duality and Hopf-Lax semi-group, [6]) For the bounded continuous function $f \in C_b$ and probability measure $\nu_{n_1}, \nu_{n_2}$ satisfying assumption **H4**, the following equality holds:*

$$\frac{1}{2}\mathcal{W}_2^2(\nu_{n_1}, \nu_{n_2}) = \sup_{\zeta \in C_b} \left[ \int H_1 \zeta d\nu_{n_1} - \int \zeta d\nu_{n_2} \right], \tag{21}$$

*where $H_t(x) = \inf_y \{\zeta(y) + \frac{1}{2t}d^2(x, y)\}$ is called a Hopf-Lax semi-group.*

Now we continue with the second step. By replacing $\zeta \to H_1 \zeta$ and subtracting $\int \zeta \mathcal{N}_\Sigma$ from both sides, we get

$$\int H_1 \zeta q_k d\mathcal{N}_\Sigma - \int \zeta d\mathcal{N}_\Sigma \leq \int H_1 \zeta P_k q_0 d\mathcal{N}_\Sigma - \int \zeta d\mathcal{N}_\Sigma + \epsilon(k). \tag{22}$$

Calculating $\sup_\zeta$ for both sides yields the following inequality according to Proposition A and assumption **H4** (finite second moments),

$$\mathcal{W}_2^2(\nu_k, \mathcal{N}_\Sigma) \leq \mathcal{W}_2^2(\mu_k, \mathcal{N}_\Sigma) + \epsilon(k). \tag{23}$$

**Third step:** For the final step, we demonstrate the exponential decay of the Wasserstein distance.

$$\begin{aligned}
\int H_1 \zeta P_k q_0 d\mathcal{N}_\Sigma - \int \zeta d\mathcal{N}_\Sigma &\leq \int P_k H_1 \zeta q_0 d\mathcal{N}_\Sigma - \int \zeta d\mathcal{N}_\Sigma \\
&\leq \int H_{e^{2\rho k}} P_k \zeta q_0 d\mathcal{N}_\Sigma - \int \zeta d\mathcal{N}_\Sigma \\
&\leq e^{-2\rho k} \left[ \int H_1 e^{2\rho k} P_k \zeta q_0 d\mathcal{N}_\Sigma - \int e^{2\rho k} P_k \zeta d\mathcal{N}_\Sigma \right],
\end{aligned} \tag{24}$$

where the first inequality is induced by the commutation properties of $P_k$ and $H_t$, and the second inequality is induced by the homogeneity of the Hopf-lax semi-group, denoted as $H_{e^{2\rho k}} P_k \zeta = e^{-2\rho k} H_1 e^{2\rho k} P_k \zeta$. The third inequality is induced as $\int P_k \zeta d\mathcal{N}_\Sigma = \int \zeta d\mathcal{N}_\Sigma$ because the target measure $\mathcal{N}_\Sigma$ is a steady-state measure of $\lim_{k \to \infty} P_k \zeta$ (invariant measure w.r.t. $P_k$). According to the arbitrariness of $P_k \zeta \in C_0^\infty$, we can obtain the following inequality by calculating $\sup_h$ for both sides of the inequality above:

$$\textbf{LHS} \longrightarrow \sup_\zeta \left[ \int H_1 \zeta d\mu_k - \int \zeta d\mathcal{N}_\Sigma \right] = \frac{1}{2}\mathcal{W}_2^2(\nu_k, \mathcal{N}_\Sigma), \tag{25}$$

and

$$\textbf{RHS} \longrightarrow e^{-2\rho k} \sup_{\zeta^*} \left[ \int H_1 \zeta^* d\nu_0 - \int \zeta^* d\mathcal{N}_\Sigma \right] = \frac{e^{-2\rho k}}{2}\mathcal{W}_2^2(\nu_0, \mathcal{N}_\Sigma), \tag{26}$$

where $\zeta^* = e^{2\rho k} P_k \zeta$. Therefore, based on the results of the previous proposition, we have

$$\mathcal{W}_2^2(\nu_k, \mathcal{N}_\Sigma) \leq e^{-2\rho k} \mathcal{W}_2^2(\nu_0, \mathcal{N}_\Sigma) \leq \rho^{-2} e^{-2\rho k} \delta_0. \tag{27}$$

By combining the above inequality with (23) and setting $k = \tau_k \in \tau(k)$, we obtain the following inequality:

$$\mathcal{W}_2(\nu_{\tau_k}, \mathcal{N}_\Sigma) \leq \sqrt{\rho^{-2} \delta_0 e^{-2\rho \tau_k} + \epsilon(\tau_k)} \tag{28}$$

The proof is completed by setting $\rho = 1$.

$\square$

### 3.3 Proofs of Perturbation Analysis

**Proposition 4.** *(Wasserstein Perturbation) Let $\nu_k^\varepsilon = (\mathbf{Id} + \varepsilon h)_\# \nu_k$ be a perturbed measure by $\varepsilon h$. Then, there are some numerical constants $0 \leq \kappa_1^1, \kappa_2^\infty < \infty$ such that the mean radius of the perturbed Wasserstein ambiguity set $\nu_k^\varepsilon \in B_{\mathcal{W}_2}(\mathcal{N}_\Sigma, r')$ is bonded as follows:*

$$\mathbb{E}_\varepsilon \mathcal{W}_2(\nu_k^\varepsilon, \mathcal{N}_\Sigma) \leq \rho^{-1} \left( \sqrt{d\kappa_1^1 \kappa_2^\infty \mathbb{E}[\varepsilon]} + \delta_k \right). \tag{29}$$

*Proof.* By the definition of the push-forward measure $\nu^{\varepsilon}$, we obtain the equality $\int g(y)d\nu^{\varepsilon} = \int g(y + \varepsilon h(y))d\nu(y) = \int g_h d\nu(y)$ with $g \in C_0^{\infty}$. By the definition of the diffusion generator $\mathcal{L}$, we obtain followings:

$$
\begin{aligned}
\sup \int \mathcal{L}^+ g(y) d\nu^{\varepsilon}(y) &= \sup \int \mathcal{L}^+ g(y + \varepsilon h(y)) d\nu(y) \\
&= \sup \int \left| \mathbf{Tr}(\Sigma \nabla^2 g_h - [y + \varepsilon h(y)]^T \nabla g_h \right| d\nu(y) \\
&\leq \sup \left[ \int \mathcal{L}^+ g_h(y) d\nu(y) + \varepsilon \int \left| h^T(y) \nabla g_h(y) \right| d\nu(y) \right] \\
&\leq \sup \int \mathcal{L}^+ g(y) d\nu(y) + \varepsilon \sup \int \sum_i^d \left| h_i \partial^i g_{i,h} \right| d\nu,
\end{aligned}
\tag{30}
$$

where $g_h(y) = g(y + \varepsilon h(y))$. For an arbitrary $a, b > 0$, the inequality $|b| - |a| \leq |a - b| \leq C$ is satisfied and $\sup|a| - \sup|b| \leq \sup(|a - b|) \leq \sup C$. By calculating the supremum of both sides of the inequality using a non-negative perturbation function $h \geq 0$ (*i.e.*, $\nu$ and the test function lying in $C_0^{\infty}$), the following expressions are induced:

$$
\begin{aligned}
\varepsilon \sup \int \sum_i^d \left| h_i \partial^i g_{i,h} \right| d\nu &\leq \varepsilon \sup \sum_i^d \int \left| h_i \partial^i g_{h,i} \right| d\nu \leq \varepsilon \sup \sum_i^d \left\| h_i \right\|_{b_1,\nu} \left\| \partial^i g_{h,i} \right\|_{b_2,\nu} \\
&\leq \varepsilon \left\| h^\star \right\|_{b_1,\nu} \sup \sum_i^d \left\| \partial^i g_{h,i} \right\|_{b2,\nu} \leq d\varepsilon \left\| h^\star \right\|_{b_1,\nu} \sup_g \max_i \left\| \partial^i g_i \right\|_{b_2,\nu^{\varepsilon}} \\
&\leq d\varepsilon \kappa_1^{b_1} \max_i \left\| \partial^i g_i \right\|_{b_2,\nu^{\varepsilon}},
\end{aligned}
\tag{31}
$$

where $\left\| h_i \right\|_{b_1,\nu} \leq \left\| h_j \right\|_{b_1,\nu} = \max_i \left\| h_i \right\|_{b_1,\nu} = \kappa_1^{b_1}$. The second inequality is induced by Hölder's inequality with the conjugates $b_1^{-1} + b_2^{-1} = 1$. Because we parameterize the test function $g$ using the smoothed function introduced in Section 2.1, we consider smoothed neural networks as members of the set of functions $A_g = \{g_\psi : g_\psi^\vartheta(y) = g^\vartheta(y) * \psi_\epsilon(y), \mathbb{E}_{\nu_k} [|\mathcal{L}g_\psi|] < \delta_k, \vartheta \in \mathbb{R}^F, \epsilon > 0\} \subset C_0^{\infty}$. It should be noted that for any member $g_\psi \in A_g$, $\partial^i g_{\psi,i} = g_i * \partial^i \psi_\epsilon \in C_0^{\infty}$. The network capacity of $g$ is sufficiently large (*i.e.*, $F$) to ensure that $A_g$ completely contains the set of functions $B = \{\phi : \mathbb{E}_{\nu_k} [\mathcal{L}[-\log \phi]] \leq \delta_k\} \subset C_0^{\infty}$, where $\int \phi d\nu_k = 1, \phi \geq 0$ a.e. $[\nu_k]$. In this case, $B \subset A_g \subset C_0^{\infty}$, and the last inequality in Proposition 2 is valid. Additionally, $\sup_B \chi \leq \sup_{A_g} \chi$ for any $\chi \in C_0^{\infty}$. Therefore,

$$
\begin{aligned}
0 \leq d\varepsilon \kappa_1^{b_1} \sup_{g \in B} \left( \int |\partial^j g_{\psi,j}|^{b_2} d\nu^{\varepsilon} \right)^{\frac{1}{b_2}} &\leq d\varepsilon \kappa_1^{b_1} \sup_{g \in A_g} \left( \int |\partial^j g_{\psi,j}|^{b_2} d\nu^{\varepsilon} \right)^{\frac{1}{b_2}} \\
&= d\varepsilon \kappa_1^{b_1} \sup_\vartheta \left( \int |\partial^j g_{\psi,j}^\vartheta|^{b_2} d\nu^{\varepsilon} \right)^{\frac{1}{b_2}} \\
&\leq d\varepsilon \kappa_1^1 \kappa_2^{\infty},
\end{aligned}
\tag{32}
$$

where $\sup \left\| \partial^i g_i \right\|_{b_2,\nu^{\varepsilon}} \leq \sup \left\| \partial^j g_j \right\|_{b_2,\nu^{\varepsilon}} = \sup \max_i \left\| \partial^i g_i \right\|_{b_2,\nu^{\varepsilon}} = \kappa_2^{b_2}$. The last inequality is induced by setting $b_1 = 1, b_2 = \infty$. Now, we rearrange the inequalities in (32) and (30) and consider the last inequality developed in Proposition 2.

$$
\begin{aligned}
\mathbb{E}_{\varepsilon} \mathcal{W}_2(\nu^{\varepsilon}, \mathcal{N}_{\Sigma}) &\leq \mathbb{E}_{\varepsilon} \int_0^{\infty} \sqrt{e^{-2\rho t}} dt \sqrt{\sup \int \mathcal{L}^+ \phi d\nu^{\varepsilon}} \leq \frac{1}{\rho} \mathbb{E}_{\varepsilon} \left( \sqrt{\sup_{\phi \in A_g} \int \mathcal{L}^+ \phi d\nu + d\varepsilon \kappa_1^1 \kappa_2^{\infty}} \right) \\
&\leq \rho^{-1} \mathbb{E}_{\varepsilon \sim p_\varepsilon} \left( \sqrt{d\varepsilon \kappa_1^1 \kappa_2^{\infty} + \delta_k} \right) \leq \rho^{-1} \left( \sqrt{d\kappa_1^1 \kappa_2^{\infty} \mathbb{E}[\varepsilon] + \delta_k} \right).
\end{aligned}
\tag{33}
$$

It should be noted that in the second inequality, the supremum is calculated for set $A_g$, which still produces the same result as $C_0^{\infty}$ according to the assumption that $B \subset A_g \subset C_0^{\infty}$. The last inequality

holds according to Jensen's inequality for concave square root functions. The proof is completed by rewriting $\kappa_1^1, \kappa_2^\infty$ as $\kappa_1, \kappa_2$ and setting $\rho = 1$. $\qquad\square$

The next corollary shows the extension of Proposition 3 to perturbed measures.

**Corollary A.** *Let $\tau_k \in \tau(k)$ be the sub-sequence defined in Proposition 3 and let $\nu_{\tau_k}^\varepsilon$ be a perturbed measure with respect to $\tau_k$. Then,*

$$\mathbb{E}_\varepsilon \mathcal{W}_2(\nu_{\tau_k}^\varepsilon, \mathcal{N}_\Sigma) \leq \frac{1}{\rho} e^{-\rho k} \sqrt{d\kappa_1^1 \kappa_2^\infty \mathbb{E}[\varepsilon] + \delta_{k=0}} + \epsilon(k). \tag{34}$$

*Proof.* This inequality is trivial to obtain by combining the results obtained in Propositions 3 and 4.

$$\mathbb{E}_\varepsilon \mathcal{W}_2(\nu_{\tau_k}^\varepsilon, \mathcal{N}_\Sigma) \leq \frac{1}{\rho} e^{-\rho k} \mathcal{W}_2(\nu_0^\varepsilon, \mathcal{N}_\Sigma) \leq \frac{1}{\rho} e^{-\rho k} \sqrt{d\kappa_1^1 \kappa_2^\infty \mathbb{E}[\varepsilon] + \delta_{k=0}} + \epsilon(k). \tag{35}$$

$\qquad\square$

**Corollary 1.** *(**Perturbed Binary Classification**.) Let $\Sigma_+$ and $\Sigma_-$ be a $r$-rank SPD matrices, and $\varepsilon \sim p_\varepsilon = \exp(b)$ be an exponential distribution with parameter $b$. Then, the probability of $\nu^\varepsilon$ classified as positive labels is bounded as follows:*

$$\mathbb{P}[\mathbf{cls}(\nu^\varepsilon) = 1] \leq 1 - e^{-\frac{br(\lambda_{max}^+ + \lambda_{max}^-) - 4b\delta_k}{4d\kappa_1 \kappa_2}}, \tag{36}$$

*where $\lambda_{\max}^+$ and $\lambda_{\max}^-$ denote maximum eigenvalues of matrices $\Sigma_+$ and $\Sigma_-$, respectively.*

*Proof.* For binary classification, we first define the decision boundary in the 2-Wasserstein distance.

$$\mathbf{D}^\varepsilon = \{\nu^\varepsilon \in \mathcal{P}_2; \mathcal{W}_2(\nu^\varepsilon, \mathcal{N}_{\Sigma_+}) = \mathcal{W}_2(\nu^\varepsilon, \mathcal{N}_{\Sigma_-})\}. \tag{37}$$

Suppose the probability measures $\xi \in \mathbf{D}^\varepsilon, \mathcal{N}_\mathbf{D}$ satisfy $\mathcal{N}_\mathbf{D} = \arg\min_\xi \mathcal{W}_2(\mathcal{N}_{\Sigma_+}, \xi)$. If the following inequality holds for any perturbed measure $\nu^\varepsilon$, then $\nu^\varepsilon$ is classified as positive label.

$$\mathcal{W}_2(\mathcal{N}_{\Sigma_+}, \nu^\varepsilon) \leq \min_\xi \mathcal{W}_2(\mathcal{N}_{\Sigma_+}, \xi) = \mathcal{W}_2(\mathcal{N}_{\Sigma_+}, \mathcal{N}_\mathbf{D}) \tag{38}$$

As our target measure $\mathcal{N}_{\Sigma_+}$ is an element of the Wasserstein Gaussian subspace $\mathcal{W}_{2,g}$ and the subspace is totally geodesic, the 2-Wasserstein distance $\mathcal{W}_2(\mathcal{N}_{\Sigma_+}, \xi)$ is minimized only when $\mathcal{N}_\mathbf{D}$ is member of $\mathcal{W}_{2,g}$ such that $\mathcal{N}_\mathbf{D} = \gamma_{0.5}$, $\gamma_t$ is geodesic connecting $\mathcal{N}_{\Sigma_+}$, and $\mathcal{N}_{\Sigma_+}$. Thus, covariance matrix $\mathbf{D}$ have unique form $\mathbf{D} = 0.25 (\mathbf{I}_d + T) \Sigma_+ (\mathbf{I}_d + T)$, where matrix $T$ is a solution of the Riccati equation $T\Sigma_+ T = \Sigma_-$. In this case, the boundary of Wasserstein ambiguity set for $\nu^\varepsilon$ exactly touches single point of the subset $\mathbf{D}_g^\varepsilon = \mathbf{D}^\varepsilon \cap \mathcal{P}_{2,g} = \{\mathcal{N}_{\Sigma_\mathbf{D}}\}$ to make $\nu^\varepsilon$ classified as positive label. In this light of consideration, the following condition is required:

$$\mathcal{W}_2(\nu^\varepsilon, \mathcal{N}_{\Sigma_+}) \leq \sqrt{d\kappa_1 \kappa_2 \varepsilon + \delta_k} \leq \mathcal{W}_2(\gamma_0, \gamma_{0.5}) = \frac{1}{2} \mathcal{W}_{2,g}(\mathcal{N}_{\Sigma_+}, \mathcal{N}_{\Sigma_-}), \tag{39}$$

where the label measure $\nu^\varepsilon$ satisfies $\nu^\varepsilon \in \mathbf{B}_{\mathcal{W}_2}(\sqrt{d\kappa_1 \kappa_2 \varepsilon + \delta_k})$ by the assumption (**H2**), and Proposition 4. By rearranging the inequality above, we obtain followings:

$$\varepsilon \leq \frac{\mathcal{W}_{2,g}^2(\mathcal{N}_{\Sigma_+}, \mathcal{N}_{\Sigma_-}) - 4\delta_k}{4d\kappa_1 \kappa_2} \leq \frac{r(\lambda_{max}^+ + \lambda_{max}^-) - 4\delta_k}{4d\kappa_1 \kappa_2}. \tag{40}$$

The inequality is induced by $r$-rank condition of covariance matrices as follows:

$$\mathcal{W}_{2,g}(\mathcal{N}_{\Sigma_+}, \mathcal{N}_{\Sigma_-}) = \mathbf{Tr}(\Sigma_+ + \Sigma_- - 2\sqrt{\Sigma_+ \Sigma_-}) \leq r(\lambda_{max}^+ + \lambda_{max}^-). \tag{41}$$

For the exponential distribution $\varepsilon \sim p_\varepsilon = \exp(b)$, we can obtain the probability inequality.

$$\mathbb{P}[\mathbf{cls}(\nu^\varepsilon) = 1] = \mathbb{P}\left[\epsilon \leq \frac{r(\lambda_{max}^+ + \lambda_{max}^-) - 4\delta_k}{4d\kappa_1 \kappa_2}\right] \leq 1 - e^{-\frac{br(\lambda_{max}^+ + \lambda_{max}^-) - 4b\delta_k}{4d\kappa_1 \kappa_2}}. \tag{42}$$

$\qquad\square$

**Proposition 5.** *(Markov Inequality for Perturbation Functions) Let $Y_k \sim \nu_k$ denote the Markov-process related to the Markov semi-group and its corresponding law $\nu_k$. For the $l$-th component of the perturbation function $h_l \in \mathbf{L}^1(\nu_k)$, we denote $T(y) = \|h(y)\|_2^2 < \infty$. Then, there are numerical constants $0 \leq \kappa_3, \kappa_4 < \infty$ such that*

$$\nu_k\big(\mathbb{E}_y[T(Y_k)] \geq a\big) \leq \frac{C(k)\kappa_3}{a^2}\left(d\kappa_4 + k\delta_k\right), \tag{43}$$

*where $C(k) = e^{\frac{1}{2(e^{2k}-1)} + 2\varepsilon^2}$, $d_M(y) = \sqrt{h^T(y)\Sigma h(y)}$, and $y \in \mathbb{R}^d$ denotes the Mahalanobis norm of $h(y)$. Furthermore, $\lim_{k\to\infty} \nu_k(\mathbb{E}[T(Y_k)] \geq a) \to a^{-2}e^{2\varepsilon^2}d\kappa_3\kappa_4$.*

*Proof.* We write the Markov inequality for the Markov semi-group $P_t T$ as follows:

$$\begin{aligned}
\nu_k(P_t T \geq a) &\leq \frac{1}{a^\alpha}\int_{\{y; P_t T(y) \geq a\}} (P_t T(y))^\alpha \, d\nu_k(y)\\
&= \frac{1}{a^\alpha}\int_{\{y; P_t T(y) \geq a\}\times\mathbb{R}^d} (P_t T(y))^\alpha \, \mathbf{1}(z) d[\nu_k(y) \otimes \nu_k^\varepsilon(z)] \\
&\leq \frac{1}{a^\alpha}\int_{\mathbb{R}^d \times \mathbb{R}^d} (P_t T(y))^\alpha \, \mathbf{1}(z) d[\nu_k(y) \otimes \nu_k^\varepsilon(z)],
\end{aligned} \tag{44}$$

where the equality holds according to the definition of the product measure. For the rest of this proof, we omit the integral area $\mathbb{R}^d \times \mathbb{R}^d$. It should be noted that $P_t T(y) \geq 0$ for all $t, y$. Now, we introduce another useful inequality:

**Proposition B.** *(Harnack's inequality [7]) If the curvature condition $CD(\rho, \infty)$ holds, then the following inequality holds:*

$$(P_t f)^\alpha(x) \leq P_t(f^\alpha)(y)e^{\frac{\alpha\rho d_G^2(x,y)}{2(\alpha-1)(e^{2\rho t}-1)}}, \tag{45}$$

*where $d_G$ is the Riemannian distance, $f$ is a positive measurable function in $\mathbb{R}^d$, and every $x, y \in \mathbb{R}^d, \alpha > 1, t > 0$.*

The first inequality is induced by Proposition B, as $T, P_t T \geq 0$ a.e. $[\nu_k]$.

$$\begin{aligned}
\int (P_t T)^\alpha(y)\mathbf{1}(z)d[\nu_k(y) \otimes \nu_k^\varepsilon(z)] &\leq \int P_t(T^\alpha)(y)e^{\frac{\alpha\rho d_G^2(y,z)}{2(\alpha-1)(e^{2\rho t}-1)}}\mathbf{1}(z)d[\nu_k(y) \otimes \nu_k^\varepsilon(z)] \\
&= \int P_t(T^\alpha)(y)\mathbf{1}(y')e^{\frac{\alpha\rho d_G^2(y,y'+\varepsilon h(y'))}{2(\alpha-1)(e^{2\rho t}-1)}}d[\nu_k(y) \otimes \nu_k(y')] \\
&\leq e^{\frac{\alpha\rho}{2(\alpha-1)(e^{2\rho t}-1)}}\|P_t T\|_{b_m, \nu_k}\left\|e^{d_G^2(y,y'+\varepsilon h(y'))}\right\|_{b_n, \nu_k \otimes \nu_k},
\end{aligned} \tag{46}$$

where the second inequality is induced because $\mathbb{E}_{y,z}[P_t T(y)\mathbf{1}(z)] = \mathbb{E}_y[P_t T(y)]\mathbb{E}_{y'}[\mathbf{1}(y' + \varepsilon h(y'))] = \mathbb{E}_y[P_t T(y)]$ and the Hölder's inequality with constants $b_m, b_n$ satisfies $b_m^{-1} + b_n^{-1} = 1$. As we discuss in Section 1.2, our diffusion operator $\mathcal{L}$ can be considered as the Riemannian diffusion on a flat manifold $(\mathbb{R}^d, d_G)$ with the Mahalanobis distance $d_G$, which has the form of $d_G(y, y') = \sqrt{(y-y')^T\Sigma^{-1}(y-y')}$ for $y, y' \in \mathbb{R}^d$. Due to the flatness of the manifold, we can easily induce the following equality:

$$\begin{aligned}
\left\|e^{d_G^2(y,y'+\varepsilon h(y'))}\right\|_{b_n, \nu_k^2} &= e^2\left|\left\|e^{d_G^2(0,y)}\right\|_{b_n, \nu_k} - \left\|e^{y^T\Sigma y'}\right\|_{b_n, \nu_k^2}\right| + e^{\varepsilon^2}\left\|e^{d_G^2(h(y), h(y'))}\right\|_{b_n, \nu_k^2} \\
&\leq e^2\left|\left\|e^{d_G^2(0,y)}\right\|_{b_n, \nu_k} - \left\|e^{y^T\Sigma y'}\right\|_{b_n, \nu_k^2}\right| + e^{2\varepsilon^2}\left\|e^{d_G^2(0, h(y))}\right\|_{b_n, \nu_k} \\
&\leq e^2\left|\left\|e^{d_G^2(0,y)}\right\|_{b_n, \nu_k} - \left\|e^{y^T\Sigma y'}\right\|_{b_n, \nu_k^2}\right| + e^{2\varepsilon^2}\kappa_3^{b_n} \approx e^{2\varepsilon^2}\kappa_3^{b_n},
\end{aligned} \tag{47}$$

where the equality holds according to simple calculations and the inequality is induced by the properties of the distance $d_G^2(0, h(y')) + d_G^2(h(y), 0) \geq d_G^2(h(y), h(y'))$. For a large perturbation

$\varepsilon \gg \sqrt{2}$, the first term on the right side of (47) is negligible, compared to second term. Based on the assumption of a constant $\kappa_3^{b_n}$, we can conclude that $\sup_k \left\| e^{d_G^2(y, y' + \varepsilon h(y'))} \right\|_{b_n, \nu_k^2} \leq e^{2\varepsilon^2} \kappa_3^{b_n}$ for a large perturbation. Next, we decompose $\int P_t T^\alpha d\nu_k$, which is related to the diffusion sequence $\delta_k$ defined earlier. According to the definition of the Markov semi-group for this type of diffusion, the first equality holds:

$$
\int P_t T^\alpha d\nu_k = \int T^\alpha d\nu_k(y) + \int \int_0^t P_s \mathcal{L} T^\alpha(y) ds d\nu_k(y)
$$
$$
\leq \int T^\alpha d\nu_k(y) + \int \int_0^t P_s \left| \mathcal{L} T^\alpha(y) \right| ds d\nu_k(y) \tag{48}
$$
$$
= \int T^\alpha d\nu_k(y) + \int_0^t \left[ \int P_s \mathcal{L}^+ T^\alpha(y) d\nu_k(y) \right] ds,
$$

where $\mathcal{L}^+ T^\alpha(y) = |\mathcal{L} T^\alpha(y)|$ is non-negative and measurable. The second equality is induced by Tonelli's theorem. By applying Tonelli's theorem again to the last term in (48), we obtain the following equality:

$$
\int_0^t \int P_s \mathcal{L}^+ T^\alpha(y) ds = \int_0^t \int \underbrace{\int \mathcal{L}^+ T^\alpha \left( e^{-s} y + \sqrt{1 - e^{-2s}} \Sigma^{\frac{1}{2}} z \right) d\nu_k(y)}_{\leq \sup_g \int \mathcal{L}^+ g d\nu_k = \delta_k} d\mathcal{N}_\Sigma(z) ds \leq t \delta_k.
$$
$$\tag{49}$$

Then, we combine the inequalities of (49) and (48). Finally, we obtain the following inequality:

$$
\| P_t T^\alpha \|_{1, \nu_k} \leq \| T^\alpha \|_{1, \nu_k} + t \delta_k. \tag{50}
$$

Next, we set the constants $b_m = 1$, $b_n = \infty$, $\rho = 1$, and $\alpha = 2$ in (46). By combining the inequalities in (46) and (50), we obtain the followings:

$$
\nu_k(P_k T \geq a) \leq \frac{1}{a^2} e^{\frac{\alpha \rho}{2(\alpha - 1)(e^{2\rho k} - 1)}} e^{2\varepsilon^2} \kappa_3^\infty (\mathbb{E}_{\nu_k} T + k \delta_k)
$$
$$
\leq \frac{1}{a^2} e^{\frac{1}{(e^{2k} - 1)} + 2\varepsilon^2} \kappa_3^\infty (d\kappa_4 + k \delta_k), \tag{51}
$$

where the last inequality is induced by the following inequality:

$$
\mathbb{E}_{y \sim \nu_k}[T(y)] = \int \| h(y) \|_2^2 d\nu_k(y) = \int \sum_l^d h_l^2(y) d\nu_k(y) \leq d \max_j \left\| h_j^2 \right\|_{1, \nu_k} \leq d\kappa_4, \tag{52}
$$

where $\max_j \left\| h_j^2 \right\|_{1, \nu_k} \leq \kappa_4$. Because we assume that the density of $\nu_k$ follows the path related to the diffusion semi-group generated by $\mathcal{L}$, we can replace the auxiliary variable $t$ with $k \in N^+$. It should be noted that $\nu_k(P_t T(y) > a) = \nu_k(\mathbb{E}_y[T(Y_k)|Y_0 = y] > a)$. The last statement holds because $\lim_{k \to \infty} C(k) = e^{2\varepsilon^2}$, and $\delta_k$ vanishes for large values of $k$ as $\delta_{k > K_0} = 0$ according to assumption **H2**. $\qquad \square$

## 4 Implementation Details

### 4.1 Perturbation Setup for 2D Images

To measure the robustness of the proposed and baseline methods on a 2D image classification task, we considered three possible perturbations.

- **Local Shuffle** $\{e\}$**.** For this perturbation, we set a local grid and shuffle each pixel in the grid. This perturbation was designed to verify learnability when the *connectivity* of pixels is locally collapsed, but the *distribution* of pixels is preserved. For example, $\mathbb{E}_\mu[I] - \mathbb{E}_{\mu^\varepsilon}[I^\varepsilon] \approx 0$ when the average pixel-wise $\mathbf{L}_p$ norm is large.
- **Downscaling, Rotation, and Sheering** $\{\theta, sc, sh, \epsilon\}$**.** This perturbation combines three possible sub-perturbations, namely, down scaling, rotation, and sheering. In the transformed

region, we added Gaussian noises with zero mean and a covariance $\epsilon \mathbf{I}_d$, where $\epsilon = 0.5$. Although the global information from each original image is still valid in each perturbed image, we observed a significant performance drop for the conventional CNN-based baselines (75.9% accuracy), whereas the proposed method yielded accurate classification results (93.0% accuracy).

- **Rotation and Cropping** $\{\theta, cc\}$**.** This perturbation combines cropping and rotation. The original images were first randomly rotated by angles $\theta = 2\pi$ and $\theta_2 = \pi$, and then cropped to sizes of $cc = (24, 24), cc_2 = (16, 16)$. Finally, the cropped images were rescaled to their original size $(32, 32)$. In contrast to the second perturbation, global information can be lost as a result of this perturbation because the images are cropped.

We used the official codes provided in [`Torchvision.transforms`] for scaling, cropping, rotation, down-scaling, and sheering. Perturbed samples are presented in Table 1.

The benchmark dataset CIFAR10-C [3] consists of artificially corrupted images with 19 different distortion operations such as *'jpeg compression'*, *'motion blur'*, *'contrast'*, and *'pixelate'*. We randomly selected each data from $10K$ numbers of most severe perturbed samples (severity $= 5$) for both training and test data. Unfortunately, each image can only possess a finite number of pre-fixed deterministic variations, which induces low randomness of perturbed objects. Due to the deterministic property of dataset, theoretical advantages of our method were limited and the experimental results taken by the proposed distributional realization produced a small margin. Nevertheless, the experimental results demonstrate that our method outperforms conventional deterministic models even though the perturbation is deterministic. The results of both models were reported at 50-epochs due to the fast convergence.

## 4.2 Perturbation Setup for 3D Point Clouds

To measure the robustness of the proposed and baseline methods for a 3D point cloud classification task, we considered three possible perturbations.

- **Random Sampling** $\{T\}$**.** We randomly sampled $1024T$-number points, where $T = 0.5$.

- **Jitter** $\{\epsilon\}$**.** We added Gaussian noises to every points with zero mean and a covariance $\epsilon \mathbf{I}_d$, where $\epsilon_1 = 0.35, \epsilon_2 = 0.7$ and $\epsilon_3 = 1.0$.

- **Random Rotation** $\{\theta\}$**.** Unlike for the 2D images, we considered geometric random rotations centered at the origin with an angle $\theta = 0.5\pi$.

- **Random Scaling** $\{s\}$**.** We randomly scaled the geometric coordinates $(sX, sY, sZ), s \sim \mathbf{Unif}[0, 1000]$ of each points.

Perturbed samples are presented in Table 2.

## 4.3 Network Architectures

Our method does not use any prior information during training to identify primitive objects in datasets. Therefore, in the presence of severe perturbations, convolutional blocks act on each pixel or point of an object separately to prevent undesirable interference between pixels or points. This procedure can be implemented using the **Conv1d** with a kernel size of 1. Each block has the following series of layers:

$$[\mathbf{Conv1d} \rightarrow \mathbf{InstanceNorm1d} \rightarrow \mathbf{ReLU}] \tag{53}$$

The last dimension is set to 128. For image classification, we used two Conv2d layers and seven convolutional blocks in (53). For point-cloud classification, we used 14 convolutional blocks. The adversarial network $g$ was composed of a **Conv1D** layer and **FC** layer in the following arrangement: $[\mathbf{Conv1d}(128) \rightarrow \mathbf{ReLU} \rightarrow \mathbf{FC}(1024, 1)]$. For the baseline models, we used ResNet18 and DenseNet121.

# 5 Ablation Study: Perturbation Norm

Table 1: **Perturbation Norm of a 2D Image** The norm is set to $p = 1, 2, \infty$. The first row lists the accuracies for different perturbation settings. The worst classification result is highlighted in <span style="color:red">red</span>. The second row lists the average pixel-wise $\mathbf{L}_p$ distances between original and perturbed images. The result with $p = \infty$ is scaled by $10^{-3}$. third row lists the mean differences between images. The strongest perturbation is presented in **bold** font.

| Perturbations | $\{e\}$ | $\{\theta_2, sc, sh, \epsilon\}$ | $\{\theta_2, cc\}$ | $\{\theta_2, cc_2\}$ |
|---|---|---|---|---|
| Samples |  | | | |
| DenseNet | 86.6 | 75.9 | 78.9 | **74.8** $\downarrow$ |
| DeepWDC | 95.9 | 93.0 | 92.7 | **87.7** $\downarrow$ |
| $\mathbb{E}_\mu \mathbf{L}_p(I - I^\varepsilon)$ | .123/.003/1.0 | **.371**/**.009**/**2.20** $\uparrow$ | .202/.005/.076 | .219/.005/.074 |
| $\|\mathbb{E}_\mu I - \mathbb{E}_{\mu^\varepsilon} I^\varepsilon\|$ | $\approx 0$ | **.214** $\uparrow$ | .009 | .013 |

Table 2: **Perturbation Norm of a3D Point Cloud** The norm is set to $p = 1, 2, \infty$. The first row lists the accuracies for different perturbation settings. The worst classification result is highlighted in <span style="color:red">red</span>. The second row lists the average point-wise $\mathbf{L}_p$ distances between original and perturbed images. The third row lists the mean differences between images. The strongest perturbation is presented in **bold** font.

| Perturbations | Original | $\{T, s, \epsilon\}$ | $\{T, s, \epsilon_2\}$ | $\{T, s, \epsilon_3\}$ |
|---|---|---|---|---|
| Samples |  | | | |
| DGCNN | $-$ | 83.6 | 68.0 | **54.8** $\downarrow$ |
| DeepWDC | $-$ | 94.8 | 85.3 | **71.9** $\downarrow$ |
| $\mathbb{E}_\mu \mathbf{L}_p(I - I^\varepsilon)$ | $-$ | .67K/14.57/2.43 $\uparrow$ | .76K/16.80/3.06 | **.87K**/**19.57**/**3.72** $\uparrow$ |
| $\|\mathbb{E}_\mu I - \mathbb{E}_{\mu^\varepsilon} I^\varepsilon\|$ | $-$ | .022 | .041 | **.251** $\uparrow$ |

Because our perturbation setup is different from those used in other methods for generating adversarial samples, it is desirable to calculate the $\mathbf{L}_p$ distance for each perturbed image. To calculate the average $\mathbf{L}_p$ distance between original and perturbed data in a data space, we calculate the perturbation norm as follows:

$$\mathbb{E}_{I \sim \mu} \mathbf{L}_p(I - I^\varepsilon) = \frac{1}{NHW} \sum_{n=1}^{N} \left( \sum_{l=1}^{HW} \left| I_{n,l} - I_{n,l}^\varepsilon \right|^p \right)^{\frac{1}{p}}, \quad p < \infty. \tag{54}$$

If $p = \infty$, then $|I_l - I_l^\varepsilon|$ is replaced with $\max_l |I_l - I_l^\varepsilon|$. The results of these calculations are presented in Tables 1 and 2. In a 2D classification task, Table 1 indicates that a large $\mathbf{L_p}$ distance in the pixel space does not necessarily yield inaccurate results for different types of perturbations. For example, the perturbation $\{\theta, cc_2\}$ yields the lowest accuracy for both the baseline methods and our method

even though its perturbation norm is not the largest due to the dummy Gaussian noises. To clarify the effect of various severe stochastic perturbations, we transformed primitive data using basic image (point-cloud) random transformations including rotation, scaling, sheering, and cropping, which are common in real-world environments.

## Footnotes

[1] Here, we make an assumption for $\varepsilon$, $\left| \frac{\epsilon(k)}{k} \right| \to 0$.