[Reviews · NeurIPS 2020]

Review 1

Summary and Contributions: The paper present a deep classification method tailored for dataset wherre both the inputs and the related labels are represented by probability measures. The proposed deep method consists of two networks of which the first maps the input measure on a predicted measure, and the second deep network renders thhe prediction diffusion-invariant. The final predicted label is the one associated to the closest target Gaussian measure in terms of Wassertsein distance. Theoretical analysis of the concepts of diffusion-invariance and Wasserstein perturbation of the predicted measure is proposed. Experimental evaluations on 2D image and point cloud classification tasks are conducted to demonstrate the effectiveness of the approach.

Strengths: * The paper addresses the iinteresting concept of classification problem where inputs and targets are both viewed as represented by prpbability measures. The setting is novel and the proposed derivations are theoretically and empirically supported. ¨The proposed architecture comprises two architectures: a measure to measure mapping network $f$ which realizes a push-forward operation and a prediction network $g$ which inputs the measure issued from $f$ and allows to predict the final label. * The diffusion-invariance property is appealing as it ensures that by lowering the average change in $g$ output, an upper bound on 2-Wasserstein distance between the desired target measure (set as centered Gaussian measure) and the mapped measure by $f$ is minimized, hence enabling the robustness of this mapped measure to perturbations. An explicit formulation of the diffusion operator amenable to computation in deep learning setting is described. Also a theoretical justification of the exponential decay of the diffusion-invariance term is provided. * To evaluate how close the estimated measure is to the target measure, these measures are restricted to centered Gaussian measures and a related Wasserstein distance between them is considered. Overall the prediction quality can be evaluated in a computationnally tractable way as the Bures distance. * Pertubation analysis of the Wasserstein distance with regard to pertubations on the input measures is provided and specialized for binary classification. It shows that the Wasserstein distance is biunded by a term depending on the square root of the perturbation magnitude * Empirical evaluations show the good performances of the approach over deterministic models.

Weaknesses: * Some crucial details of the derivation are skipped in the text and this does not ease the understandiung of the paper. For instance, the paper must make precise how the true labels are set as Gaussian measures. Also it should justify the use of centered Gaussian measures and specify what does it entail to consider non null means for the target measures. Other skipped details are the intuition and the justification behind the parameter setting of the diffusion operator. * In the same spirit, the main paper must highlight how the overall practical learning scheme is shaped and must include the learning algorithm.

Correctness: The empirical methodoly described in the supplemental follows the standards of machine learning. The proofs of the theoretical results were not checked thoroughly.

Clarity: * The material of the paper is dense with several notions that desserve to be defined before their use. Some examples are listed below. * Ornstein-Uhlenbeck process notion is not defined. $L_1$ is not defined in Proposition 6 which as a comment is difficult to grsp. Either the authors explin the meaningfullness of the latter propoistion or they deport it ti the supplemental. * Defining the notion of Markov semi-group before its use Line 118 would strongly improve the easy reading of the paper. * Line 220: $C_b$ is not defined Line 56: "carnality" --> "cardinality" * Line 145: set a reference for the Mehler’s formula * Line 218: "form" --> "from"

Relation to Prior Work: Relation to previous work is discussed.

Reproducibility: Yes

Additional Feedback: The provided implementaion details in the appendix may allow to attempt to reproduce the results. However they do not specify how the batch size and the way the target measures are set in practice. After rebuttal --------------- The feedback addresses raised issues according to the skipped details in the main text. The setting of the true label as Gaussian measures is clarified. So is the justification of the use of centered Gaussian measures although the benefits of considering non null means Gaussian measures is not discussed. Setting of the diffusion parameter is not clarified.


Review 2

Summary and Contributions: The paper introduces a method to tackle the classification task when the data have perturbations. The authors formulate both the samples and their labels as probability measures in the Wasserstein space and regard the classification task finding the correct functional maps. True labels are assumed to be Gaussian with different variances so that the computation of OT can be theoretically reduced and the Wasserstein distance be bounded by a diffusion operator that the authors propose. The paper discusses in length the connections of the operator to the Wasserstein distance and its properties. The paper includes some empirical results from classification on images and 3D point clouds in which the proposed method outperforms several baselines.

Strengths: -- soundness of the claims (theoretical grounding, empirical evaluation) The derivation of the theorems seem solid. -- significance and novelty of the contribution The paper formulates the classification task from a novel perspective -- relevance to the NeurIPS community. The paper seems to broaden our understanding of OT and the Wasserstein space.

Weaknesses: -- The motivation of the work is not clear to me. The only motivation-ish I can find is Line 31-41, "...perturbations are randomly added to data points... represent data using a probability measure rather than as an individual observed data point because data can have multiple locations at every observations. In this circumstance, we have to minimize the Wasserstein uncertainty of perturbed data represented as probability measures to become predictable." 1st, in Line 295, "we generated severe structural perturbations, including random rotations, random resizing, and non-homogeneous local noises." Does any of these perturbations simulates "data can have multiple locations at every observations"? 2nd, "In this circumstance, we have to minimize the Wasserstein uncertainty of perturbed data", what is the connection between Wasserstein and perturbation? Why Wasserstein? -- Experiments It would be better to test the method on data with real perturbations, e.g. noisy 3D scans, raw image captures, and shaking sensors, instead of simulated ones. And BTW I cannot think of a real scenario with existing classification datasets where "data can have multiple locations at every observations".

Correctness: I haven't found any major errors.

Clarity: The writing is satisfactory in terms of the language. However, I find the contents and the logic opaque and flowing too fast. E.g., I cannot find the connection between 2.1, 2.2, 2.3, and 2.4. How do these four components form a story? In 2.4, "To evaluate the classification performance, we propose the following... distance." Where does this distance come from? What is the rationale behind it? The authors included much basics in the supplementary including notations and some definitions. I think some of them should be induced in the main paper. It is almost impossible for readers who are not familiar with the topic to understand it, at least without checking the supplementary.

Relation to Prior Work: Yes, to the best of my knowledge.

Reproducibility: Yes

Additional Feedback: Table 2, column 2 and 3. How do we explain that DeepWDC achieves a much better result in 3 (93%) than in 2 (88.6%) but the baselines have the opposite trend which suggests 3 is harder than 2? --- After rebuttal: Unfortunately, the authors' response is not convincing to me. While the majority of the reviewers favor acceptance, I stand by my previous comments and rating (below the bar). I think the paper can be significantly improved by narrating a clear story and motivation and strengthening the evaluation part.


Review 3

Summary and Contributions: This paper presents an interesting idea of 1) treating features and labels as probabilistic measures to describe geometrically transformed input data instances (shearing/rotation/scale transformation) 2) introducing Wasserstein distance and diffusion invariance measure into the learning objective to reach transformation-robust classification. We have read through the feedbacks, which address our concerns properly. Thanks for the authors' efforts.

Strengths: It is an interesting idea to treat both feature and labels as probabilistic measures and use probabilistic distribution measures to describe the diffusion caused by geometrical transformation of the original objects. By introducing Wasserstein Gaussian Space in the objective function design, it gains good analytical property. The experimental study demonstrates the robustness of the method against severe geometrical transformations.

Weaknesses: 1. Why does the diffusion-invariant distance measure help to improve robustness to shearing / rotation / scale transformation of the original data ? It is not clearly explained. More specifically, it would be helpful to explain how image geometric transformation / distortion can be presented in drift of feature space distribution. How much distortion can be tolerated by the diffusion-invariant measurement ? Furthermore, I would assume that there is a trade-off between the invariance to image distortion and sensitivity to image content change. Would it be better if there is some discussion over the balance between the two ends. 2. Why should be the first term in Eq.4 maximized with respect to the neural network parameters ? Why does maximizing this term help to improve the diffusion-invariance ?

Correctness: Yes.

Clarity: Yes, it is well written.

Relation to Prior Work: Yes, it is clearly discussed

Reproducibility: Yes

Additional Feedback:


Review 4

Summary and Contributions: By representing input data and target labels as probability measures, this paper proposes a deep Wasserstein distributional classification method. The proposed method optimizes the diffusion invariant operator of estimated label measure together with the intrinsic distance between the estimated label measure and target label measure in the Wasserstein Gaussian subspace. Experiments have shown that the proposed method is very robust to severe perturbations in input data.

Strengths: The proposed distance-based distributional classifier is novel, different from previous deep neural network based classification framework. In addition, the authors provide in-depth theoretical analysis on the connection between the diffusion operator and 2-Wasserstein distance, and on the ability of the method to quickly reduce Wasserstein ambiguity. The proposed method is very robust to severe random perturbations, achieving big performance gains on 2D image classification and 3D point could classification.

Weaknesses: (1) As suggested by the title and abstract, the paper focuses on a distance-based classification methodology based on deep neural networks (DNN) using the idea of adversarial training. While I agree the definitions and theoretical analysis are important to underpin the proposed method, the organization and writing of the present manuscript makes it very hard to obtain the global picture of the proposed methodology. (2) Experimental results on Table 1 and Table 2 are awesome, demonstrating the proposed classification method significantly outperforms the baseline, deterministic classification methods by 11%~17% in most perturbation scenarios. Could the authors give more details on how the baseline networks are trained, and clarify the difference of training setting between the baselines and the proposed methods? On standard perturbation datasets (i.e., CIFAR 10-C), their performance gap is not that significant (i.e., 2.8%); could authors provide more analysis on this point? ---------------------------------------------------------------------------- I have read the authors' response. The authors did not address my concern about experimental setting. I am afraid that comparison in Table 1 and Table 2 to the baseline, deterministic methods may not be fair; primarily, it is not clear whether the baseline methods use the perturbed images during training as data augmentation, while the proposed method use indeed. In addition, I think that the paper's writing requires much improvement so that the readers can have a clear global picture of motivation, method and contributions. In light of the above consideration, my final recommendation is "marginally above the acceptance threshold".

Correctness: The proposed method seems to be technically correct, though I did not check completely.

Clarity: The paper is organized and written in such a way that it is hard to have a clear global picture of the proposed method.

Relation to Prior Work: Differences from previous work are clearly discussed.

Reproducibility: No

Additional Feedback:

[Author Response · NeurIPS 2020]

*Deep Diffusion-Invariant Wasserstein Distributional Classification*

We would like to thank the reviewers for their time and effort to read our paper and provide constructive suggestions. We carefully addressed all comments as closely as possible. All reviewers agree that the overall system contains a number of potentially interesting ingredients and can represent a worthy contribution. Hopefully, the paper in its present form satisfies the requirements of the reviewers. Thus, please give a chance to publish our work in NeurIPS 2020.

———————————— **Reviewer #1** ————————————

Some details were omitted in the main paper due to the lack of space, but we attempted to add as many details as possible to Appendix. According to your instruction, we will move the core definitions in Appendix to the main paper. Please appreciate in-depth and theoretical analysis of the proposed method in the main paper and Appendix.

**C1. Gaussian measures.** We represented target Gaussian measures as $r$-rank covariance matrices, $\Sigma_c = \mathbf{M}_c^T \mathbf{M}_c$, where $\mathbf{M}_c$ denotes $(r \times d)$ size of random matrix for the $c$-th class and all indices are i.i.d uniform random variables. We use the centered Gaussian measures, because it is stationary against the diffusion operator in Proposition 1. If we consider non-null means, the Mehler's formula is not explicitly defined, which is a crucial problem for efficient learning.

**C2. Details.** The batch-size was set to $128$. We will more rigorously define the notations in the camera ready paper.

———————————— **Reviewer #2** ————————————

**C1. Our motivation on the data with multiple locations.** We mean that each pixel of data can change its location, if a different perturbation is applied. For example, a single 2D image is represented as a set of the pixels, $\{x \in \mathbb{R}^3\}$, where $x$ denotes a single pixel. Then, each pixel $x$ can be contaminated into by various factors (*e.g.*, local noises and rotations), which produces multiple variants of $x$ (*e.g.*, $x + \varepsilon_1$ and $x + \varepsilon_2$), as depicted by red points in Figure 1. If $\varepsilon_1 = (-10, 0, 0)$, *i.e.*, random noise, $x$ is shifted to the left along the $x$-axis, which has a location $x + (-10, 0, 0)$. The CIFAR-10-C dataset contains the aforementioned variations of a single data. *Our goal is to cover all of these variants by finding the optimal representation in terms of the Wasserstein uncertainty and to classify them into the same class.*

**C2. Our motivation on Wasserstein uncertainty.** Our motivation is to suggest a new framework to deal with *stochastic perturbations* with tools developed in OT. The Wasserstein uncertainty corresponds to the 2-Wasserstein distance between diffusion variant (affected by perturbations) and invariant (hardly affected by perturbations) label measures. Thus, minimizing the Wasserstein uncertainty is equivalent to protecting the proposed label estimation process from several perturbations. The Wasserstein distance was used, because it can handle measures (*i.e.*, distributions).

**C3. Real-world datasets.** We have results on real-world perturbations and will include them to the final version.

**C4. Table 1.** We believe that a severe rotation setting, *i.e.*, $\theta_2 = 2\pi$, makes our diffusion classifier $g$ much easier to capture global information, in which the measure $\nu$ becomes smoother and predictable in the Wasserstein space.

———————————— **Reviewer #3** ————————————

**C1. Why diffusion-invariance help to improve robustness?** The goal of conventional classification methods is to fit the mapped data $f(x) \in \mathbb{R}^d$ to the true label $\hat{y} \in \mathbb{R}^d$ given as an one-hot vector. Contrary, the goal of our method is to fit the mapped distribution $\{f(x) \in \mathbb{R}^d\}$ to a set of i.i.d Gaussian vectors $\{Z \in \mathbb{R}^d\}$ given as a probability measure, where the Gaussian probability measure is robust to random perturbations (*i.e.*, stationary against the diffusion operator in Proposition 1). Thus, the shape of $\{f(x) \in \mathbb{R}^d\}$ should be similar to that of the target Gaussian distribution. In this context, the diffusion-invariance term in eq.(4) measures how $\{f(x) \in \mathbb{R}^d\}$ is differed from the target Gaussian distribution. If the geometric transformations are severe, the shape of $\{f(x) \in \mathbb{R}^d\}$ is largely fluctuated and is highly non-Gaussian, which makes the diffusion-invariance term to yield a large value. Then, our method tries to minimize this diffusion-invariance term, and thus can reduce the aforementioned fluctuation (*i.e.*, perturbation).

**C2. How much distortion can be tolerated?** Corollary 1 shows that the probability of accurate classification is bounded exponentially inverse to the value of $\delta$, which is the diffusion-invariance term (degree of distortion) in eq.(4).

**C3. Balance between invariance and sensitivity** Our method can balance between the invariance to image distortion (by diffusion invariance term) and sensitivity to image content change (by intrinsic distance term). We will Clarify it.

**C4. Why $g^\vartheta$ is maximized?** The choice of function $g$ in eq.(4) is crucial for the diffusion-invariance term. By maximizing $g^\vartheta$, we can find the most sensitive function which will accurately measure the diffusion-invariance.

———————————— **Reviewer #4** ————————————

We tried to describe the global picture in Figure 1 and provide details in Appendix. We will further detail our method.

**C1. Explanation on performance gap on CIFAR10-C.** The sampled images in the CIFAR-10-C only possess 19 pre-fixed deterministic variations, which induce low randomness of perturbed objects. While the main contribution of our method is to show not only the robustness to severe perturbations but also the robustness to highly stochastic perturbations, theoretical advantages developed in Section 3.3 are limited due to the low randomness, which leads a small margin in accuracy. As we mentioned above, the perturbed images are recognized as: $X^\varepsilon = \{(x + \varepsilon) \in \mathbb{R}^3\}$. In this case, the probability measure for random variable $\varepsilon$ can be defined as $\sum_{n=1}^{19} \delta_n$ for Dirac-delta measure $\delta$ supporting 19 varying images. However, this probability measure is highly concentrated and non-smooth, which can not be properly captured by our diffusion classifier $g$.

[Meta-Review · NeurIPS 2020]

This paper proposes a new classification algorithm by turning both the input data and target label into probability measures in the 2-Wasserstein space. A network is trained to push the raw input forward to the predicted measure, while another network enforces diffusion invariance. Theoretical analysis was provided for the relationship/equivalence between the diffusion operator and the 2- Wasserstein distance. Experimental results on 2D image and point cloud classification suggest the effectiveness of the method under severe random perturbation. All reviewers find the approach interesting. One reviewer, and myself to some extent, is not clear about the motivation behind Wasserstein uncertainty and the data with multiple locations. The rebuttal soothed the concern a little, but it will be helpful to better clarify the motivation. The experiment is also thin, and the reviewers (R2 and R4) still have concerns after the rebuttal. I understand the paper’s major contribution is theoretical, but it is important to show that the problem address is real by testing on real perturbations. Overall, I see the potential of the method and I encourage the reviewers to enhance the experiments in the final version where an additional page will be allowed.